# Recognition of familiar food activates feeding via an endocrine serotonin signal in *Caenorhabditis elegans*

**Bo-mi Song[1,2]\*, Serge Faumont[3], Shawn Lockery[3], Leon Avery[1]**

[1]Department of Physiology and Biophysics, Virginia Commonwealth University, Richmond, United States; [2]Department of Molecular Biology, University of Texas Southwestern Medical Center, Dallas, United States; [3]Institute of Neuroscience, University of Oregon, Eugene, United States

**Abstract** Familiarity discrimination has a significant impact on the pattern of food intake across species. However, the mechanism by which the recognition memory controls feeding is unclear. Here, we show that the nematode *Caenorhabditis elegans* forms a memory of particular foods after experience and displays behavioral plasticity, increasing the feeding response when they subsequently recognize the familiar food. We found that recognition of familiar food activates the pair of ADF chemosensory neurons, which subsequently increase serotonin release. The released serotonin activates the feeding response mainly by acting humorally and directly activates SER-7, a type 7 serotonin receptor, in MC motor neurons in the feeding organ. Our data suggest that worms sense the taste and/or smell of novel bacteria, which overrides the stimulatory effect of familiar bacteria on feeding by suppressing the activity of ADF or its upstream neurons. Our study provides insight into the mechanism by which familiarity discrimination alters behavior.

**\*For correspondence:** bomi.song@gmail.com

**Competing interests:** The authors declare that no competing interests exist

## Introduction

Wholesome food is essential for survival and animals have developed nervous systems that guide food intake. The nervous system senses diverse extrinsic and intrinsic cues, integrates the information and activates muscle movements that are required for food intake. The nervous system also stores past food experiences, which change the pattern of food intake. Dissection of the neural pathways that control food intake is not only a key to stop the epidemic of obesity and eating disorders, but may also provide insight into fundamental problems in neuroscience such as sensory perception and learning and memory.

Recognition is the ability to identify and to judge a recently encountered item as having been presented previously (*Brown and Aggleton, 2001*). In response to previously encountered stimuli, this ability allows knowledge gained from prior experience to guide animals to respond with an altered output that is beneficial for their survival. Recognition is classified into two types: recollection and familiarity discrimination. Recollection is knowledge of prior occurrence with vivid contextual details. In contrast, familiarity discrimination is mere sensation of prior occurrence and thus does not accompany episodic memory (*Brown and Aggleton, 2001*). Accumulated studies show that mere exposure to particular food alters subsequent consumption of the food in many different species (*Pliner et al., 1993*; *Wang and Provenza, 1996*; *Diaz-Cenzano and Chotro, 2010*; *Morin-Audebrand et al., 2012*), suggesting that feeding regulation by familiarity discrimination is conserved across species. Some species including humans consume familiar food more actively than novel food (*Diaz-Cenzano and Chotro, 2010*), probably to avoid possible pathogens. In contrast, other species consume familiar food less actively than novel food (*Wang and Provenza, 1996*), probably to assure balanced nutrition

**eLife digest** Many species, including our own, show a preference for familiar foods over novel ones. This behavior probably evolved to reduce the risk of consuming items that turn out to be poisonous, but the mechanisms that underlie a preference for familiar foods are largely unknown.

The nematode worm, *C. elegans*, is a useful organism in which to study such processes. Having only around 1000 cells and a simple anatomy, *C. elegans* is an attractive model system for studying molecular biology, and was the first multicellular organism to have its genome fully sequenced.

*C. elegans* feeds on bacteria, which it detects using a pair of sensory cells called ADF neurons, which sense extrinsic cues. When the ADF neurons detect bacteria, they release the transmitter serotonin. Serotonin stimulates the worm to consume the bacteria by pumping them into the pharynx, its feeding organ, and then transporting them to its intestine after crushing them.

Now, Song et al. have demonstrated that *C. elegans* consumes familiar bacteria more rapidly than it does novel ones, and have identified the molecular mechanism behind this behavior. They found that familiar bacteria stimulated the release of serotonin from the ADF cells: this activated SER-7 receptors on a specific type of motor neuron in the pharynx and this, in turn, triggered the worms' feeding response. Novel bacteria, on the other hand, failed to either activate ADF or to trigger feeding. Moreover, when Song et al. offered the worms familiar bacteria in medium that had previously contained novel bacteria, the residual traces of the novel bacteria stopped the worms from responding to familiar food.

Further research is needed to determine whether the mechanisms that underpin the more active consumption of familiar food by *C. elegans* can also explain the preference for familiar foods shown by other species. A better understanding of the mechanisms by which different foods elicit feeding could also offer important insights into factors that contribute to obesity.

intake. Despite extensive studies of recognition (*Brown and Aggleton, 2001*; *Barker et al., 2006*; *Seoane et al., 2009*; *Uslaner et al., in press*) and subsequent behavioral plasticity (*Kandel and Schwartz, 1982*; *Kravitz, 1988*), the mechanisms by which familiarity discrimination alters food intake are still poorly understood.

Its genetic tractability and simple anatomy make the bacteria-eating roundworm *C. elegans* (*Schafer, 2005*) an attractive model system for study of the process. Although it is unknown if familiarity discrimination alters food intake in *C. elegans*, the following observations support the possibility: The nervous system in *C. elegans* senses various aspects of food, such as the efficiency with which it supports growth (*Shtonda and Avery, 2006*) and its pathogenicity (*Zhang et al., 2005*), and triggers behavioral plasticity. The nervous system can also form memories of various olfactory or gustatory cues (*Bargmann, 2006*), which is likely to be crucial for the recognition of familiar food. Here, we show that *C. elegans* discriminates familiar food from novel food and selectively increases feeding in response to familiar food. Using the behavioral pattern that we identified, we uncover the mechanism by which familiarity discrimination increases the feeding response.

## Results

### Recognition of familiar bacteria increases the feeding response in *C. elegans*

To test if familiarity discrimination alters feeding in *C. elegans*, we tested if exposure to a particular bacterium alters its subsequent consumption. For this assay, we trained wild-type animals to develop familiarity either to *Escherichia coli* HB101 (H) or to *Pseudomonas* DA1878 (D) (also called B7 in our previous study, *Avery and Shtonda, 2003*) by exposing the animals to one or the other bacterium from the first larval stage (L1) until adulthood (*Figure 1A*). HB101 and DA1878 are benign bacteria that support the growth of *C. elegans* at a similar rate (*Avery and Shtonda, 2003*). Once the animals reached adulthood, we compared feeding rates on previously experienced bacteria (HH and DD groups) to feeding rates on novel bacteria (DH and HD groups) (*Figure 1A*; see 'Feeding assay' and 'Statistical analysis and data presentation' in 'Materials and methods' for details). We found that the

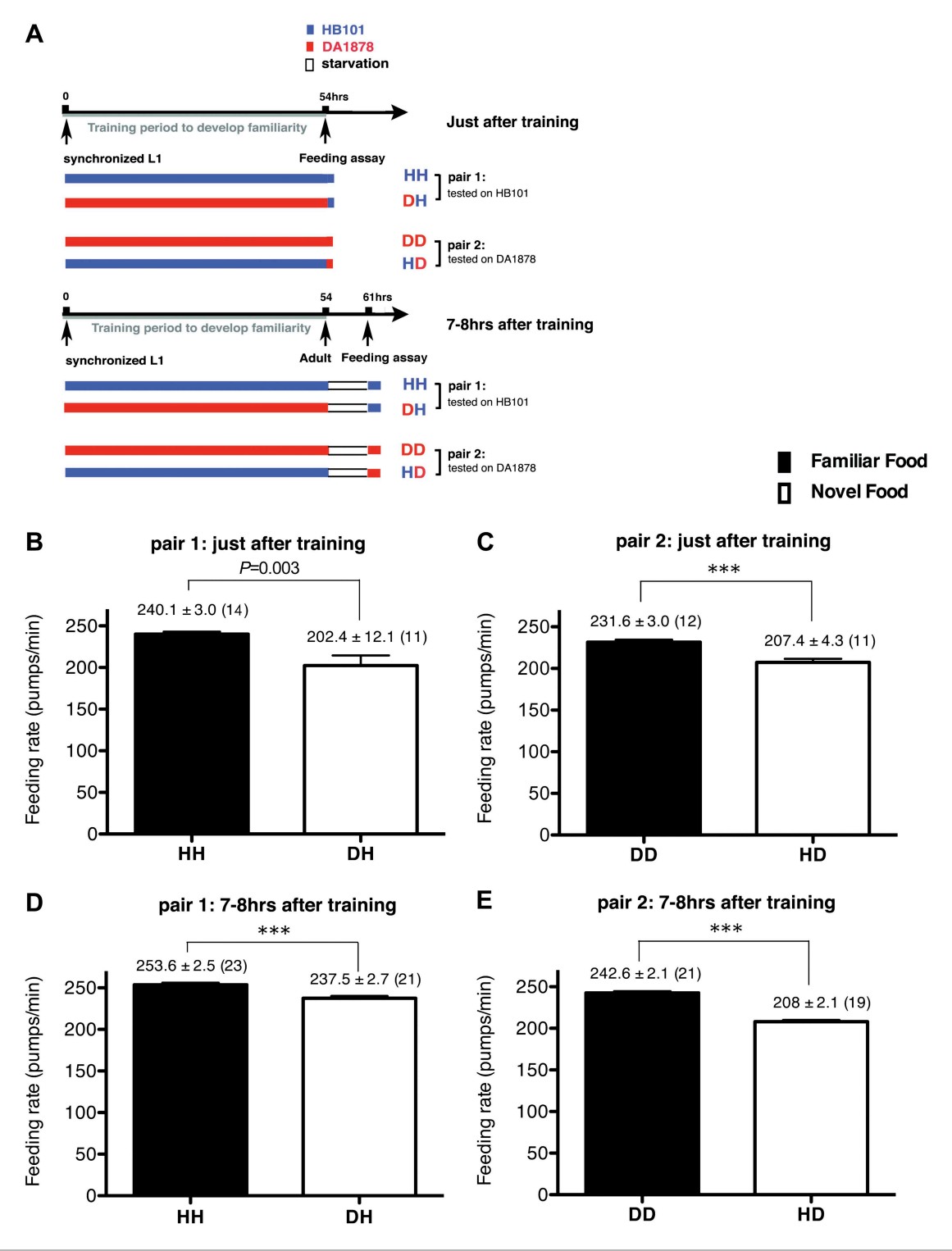

**Figure 1**. Recognition of familiar food increases feeding response in *C. elegans*. The memory of familiar food lasts for at least 7 hr. (**A**) Experimental design for the feeding assay. The periods during which animals were exposed to HB101, DA1878, and starvation are shown in blue, red, and white, respectively. Each condition is coded by two letters representing the training and test food in order. H and D represent HB101 and DA1878, respectively. (**B**)–(**C**) Feeding rates of wild-type worms on HB101 (**B**) and DA1878 (**C**) just after training the animals on one or the other bacterium. (**D**)–(**E**) Feeding rates of wild-type worms on HB101 (**D**) and DA1878 (**E**) after a 7- to 8-hr interval from training the animals on one or the other bacterium. Data shown as mean ± SEM, ***p<0.001, unpaired t-test and Mann–Whitney U test (two-tailed). The number of animals tested (n ≥ 3 independent assays for each group) is shown in parenthesis.

feeding rates of the animals on familiar bacteria were significantly higher than the rates on novel bacteria, regardless of bacterial type (*Figure 1B,C*). The increased feeding rates on familiar food compared to the rates on novel food persisted 7–8 hr after the training was over, supporting that worms discriminate familiar food from novel food and that recognition of familiar bacteria increases feeding (*Figure 1D,E*). Consistent with this, the familiar food-induced increase in feeding was not affected by the worm's nutritional status (*Figure 1B–E*). We then tested if the behavior is selective for the two tested bacteria by repeating the experiment using other benign bacterial strains, *Enterobacteria* JU54 and *Pseudomonas* PA14 *pstP* (*Tan et al., 1999*) (*Figure 2A*). Consistent with the idea that mere exposure to any benign bacterium increases subsequent consumption of that bacterium, the feeding rates on familiar bacteria were higher than the rates on novel bacteria, regardless of bacteria type (*Figure 2B–E*). Finally, we considered the possibility that cultivation on two different bacteria caused the differences in feeding rates on familiar food and novel food by affecting development. If the feeding differences are caused by developmental differences, not familiarity discrimination, an exposure to particular bacteria during adulthood is not expected to cause an increased feeding response to the bacteria. However, 9 hr of exposure to a particular bacterium during adulthood was sufficient to induce a preferential response to that bacterium (*Figure 3D,E*). In contrast to 9 hr, 6-hr exposure to the training food failed to increase subsequent consumption (*Figure 3B,C*), suggesting that the behavioral plasticity was determined by the duration of exposure to the training food. These data imply that *C. elegans* forms a recognition memory of particular bacteria after experience, which allows the worms to discriminate familiar bacteria from novel bacteria, and that the recognition of familiar bacteria increases the feeding response.

## Recognition of familiar food increases the feeding response by activating serotonin signaling via SER-7

The neurotransmitter serotonin increases feeding in *C. elegans* (*Croll, 1975*; *Avery and Horvitz, 1990*) and serotonin has long been suggested to be a food signal in *C. elegans* (*Horvitz et al., 1982*). However, the serotonin effect on feeding was tested only on familiar bacteria; thus it is unknown if the serotonin feeding signaling is activated by novel bacteria. We thus hypothesized that recognition of familiar bacteria might increase the feeding response by activating serotonin signaling. To test our hypothesis, we first tested if a *tph-1* null mutation suppresses the difference in the feeding rates between animals on familiar bacteria and novel bacteria. *tph-1* encodes a tryptophan hydroxylase required for serotonin biosynthesis (*Sze et al., 2000*). We found no difference in the feeding rates between the *tph-1* null mutant animals on familiar bacteria and novel bacteria (*Figures 4A,B and 5*). Furthermore, exogenous serotonin treatment increased the feeding rates on novel food to rates comparable to those on familiar food without affecting the rates on familiar food (*Figure 4C,D*). These data suggest that recognition of familiar bacteria indeed increased feeding rate by activating serotonin signaling. To find out which serotonin receptors mediate the serotonin action on feeding, we compared the feeding rates between wild-type and serotonin receptor null mutants in presence of serotonin. The *C. elegans* genome encodes four serotonin-activated G protein coupled receptors, SER-1, SER-4, SER-5 and SER-7, and a serotonin-gated Cl⁻ channel MOD-1. Among the tested mutants, only the *ser-7* null mutant failed to activate feeding in presence of serotonin, confirming the previous report that serotonin activates feeding via a type 7 serotonin receptor SER-7 (*Hobson et al., 2006*; *Song and Avery, 2012*) (*Figure 6*). Consistent with the idea that serotonin increases feeding via SER-7 in response to familiar food, the feeding rates of *ser-7* null mutants on familiar food were substantially decreased compared to wild type (*Figure 7A,B*). Furthermore, no differences were found in the feeding rates among the *tph-1* single null mutant, the *ser-7* single null mutant and the *tph-1; ser-7* double null mutant on familiar bacteria (*Figure 8A*). Since SER-7 is the major receptor to mediate the serotonin action, we initially expected no difference between feeding rates of the *ser-7* null mutant on familiar food and novel food, as for the *tph-1* null mutant. When the *ser-7* null mutant animals were tested on DA1878, the feeding rates of the animals did not differ on familiar and novel food (*Figures 4F and 7B,D*). However, when the animals were tested on HB101, the feeding rate of the animals was lower on familiar test food than on novel test food (*Figures 4E and 7A,C*). To resolve the unexpected observation, we examined feeding rates in wild type and the *ser-7* null mutant animals in the presence or absence of serotonin. Serotonin suppressed feeding in the *ser-7* null mutant (*Figure 4G*), suggesting that serotonin suppresses as well as activates feeding. We also found that null mutations in *ser-4* and *mod-1* completely relieved the suppression of feeding by serotonin in the

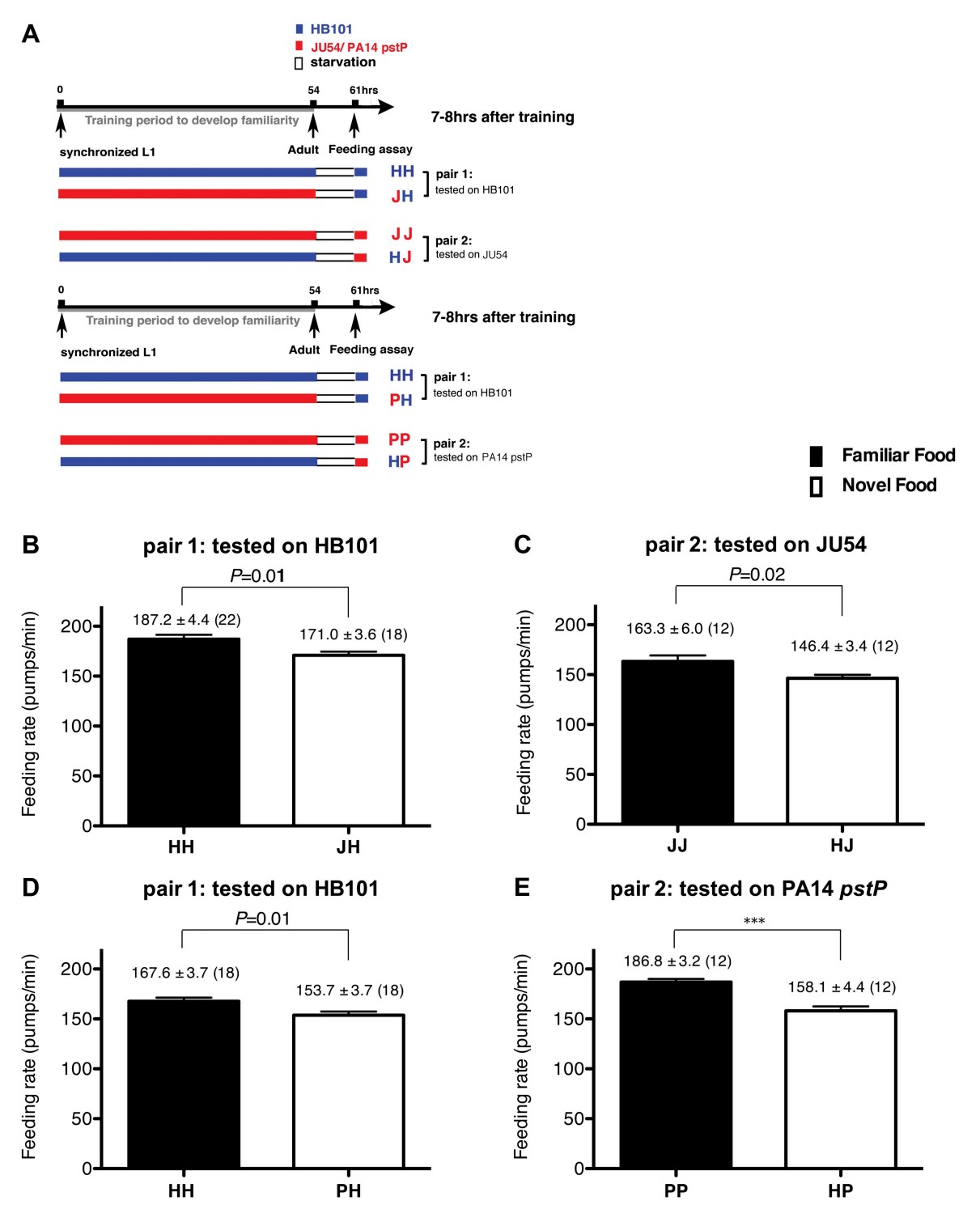

**Figure 2**. Recognition of familiar food increases feeding response in *C. elegans*. The memory of familiar food lasts for at least 7 hr. (**A**) Experimental design for the feeding assay. The periods during which animals were exposed to HB101, JU54 or PA14 *pstP*, and starvation are denoted blue, red, and white, respectively. Each condition is coded by two letters representing the training and test foods in order. H, J and P represent HB101, JU54 and PA14 *pstP*, respectively. (**B**)–(**C**) Feeding rates of wild-type worms on HB101 (**B**) and JU54 (**C**) after a 7- to 8-hr interval from training the animals on one or the other bacterium. (**D**)–(**E**) Feeding rates of wild-type worms on HB101 (**D**) and PA14 *pstP* (**E**) after a 7- to 8-hr interval from training the animals on one or the other bacterium. Data shown as mean ± SEM, ***p<0.001, unpaired t-test and Mann–Whitney U test (two-tailed). The number of animals tested (n ≥ 3 independent assays per each group) is shown in parentheses.

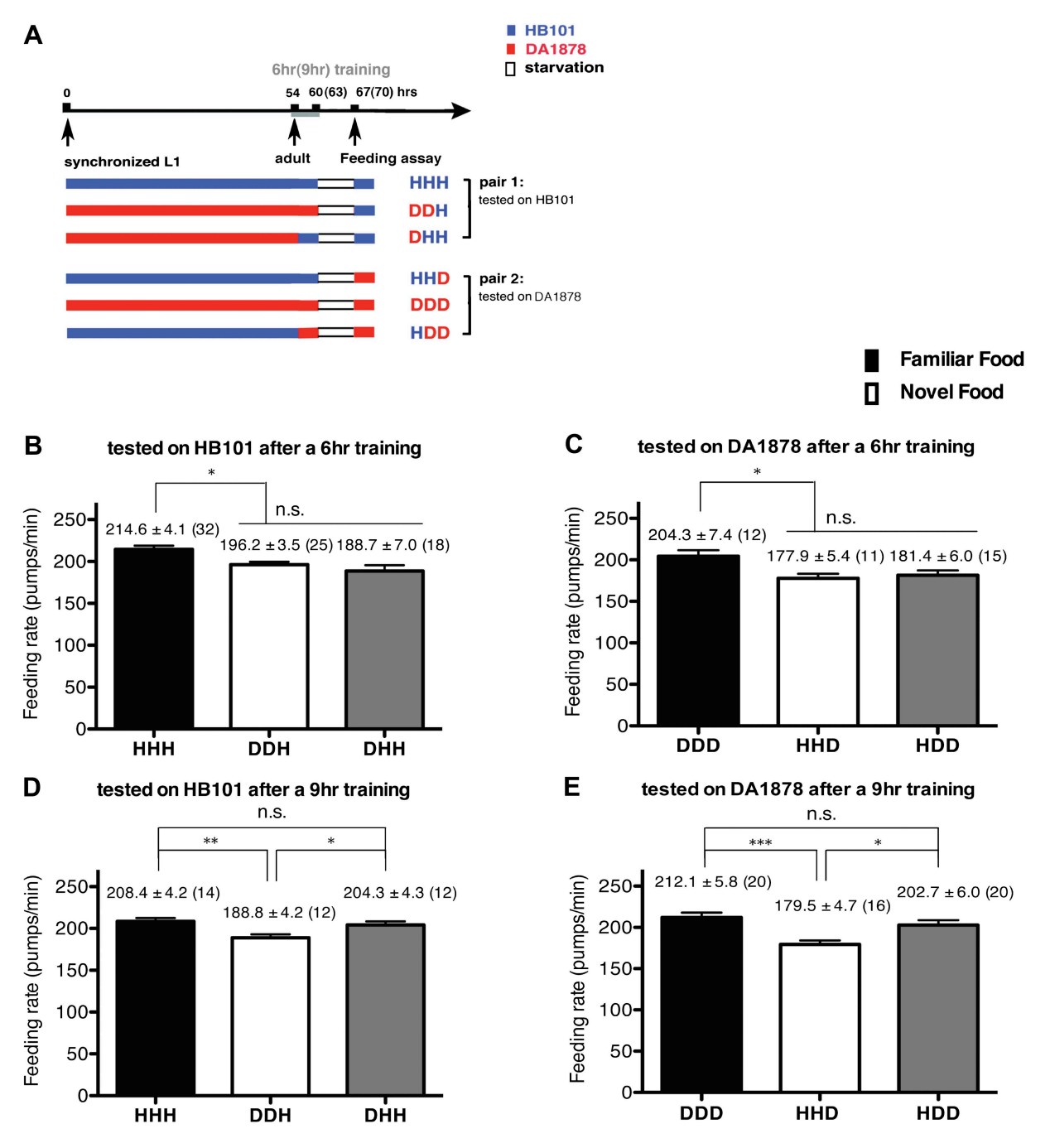

**Figure 3**. A 9-hr exposure, but not a 6-hr exposure, to particular bacteria during adulthood increases its subsequent consumption. (**A**) Experimental design for the feeding assay. Coding is as in **Figure 1A**. Each condition is coded by three letters representing the cultivation, training and test food in order. (**B**)–(**E**) feeding rates of wild-type worms on HB101 (**B** and **D**) And DA1878 (**C** and **E**) After a 7- to 8-hr interval from training the animals on one or the other bacterium. Data shown as mean ± SEM, n.s., not significant (p≥0.05), *p<0.05, **p<0.01, ***p<0.001, one-way ANOVA, post hoc Tukey test. The number of animals tested (n ≥ 3 independent assays per each group) is shown in parenthesis.

*ser-7* mutant (**Figure 4G**), suggesting that serotonin suppressed feeding by acting on SER-4 and MOD-1. In support of the idea that the inhibitory serotonin signal via SER-4 and MOD-1 is active only on familiar food, the feeding rates of the *ser-4; mod-1; ser-7* triple null mutant were greater than the rates of the *ser-7* single null mutant on familiar food (**Figures 4E,F and 7C,D**) whereas the feeding rates of the

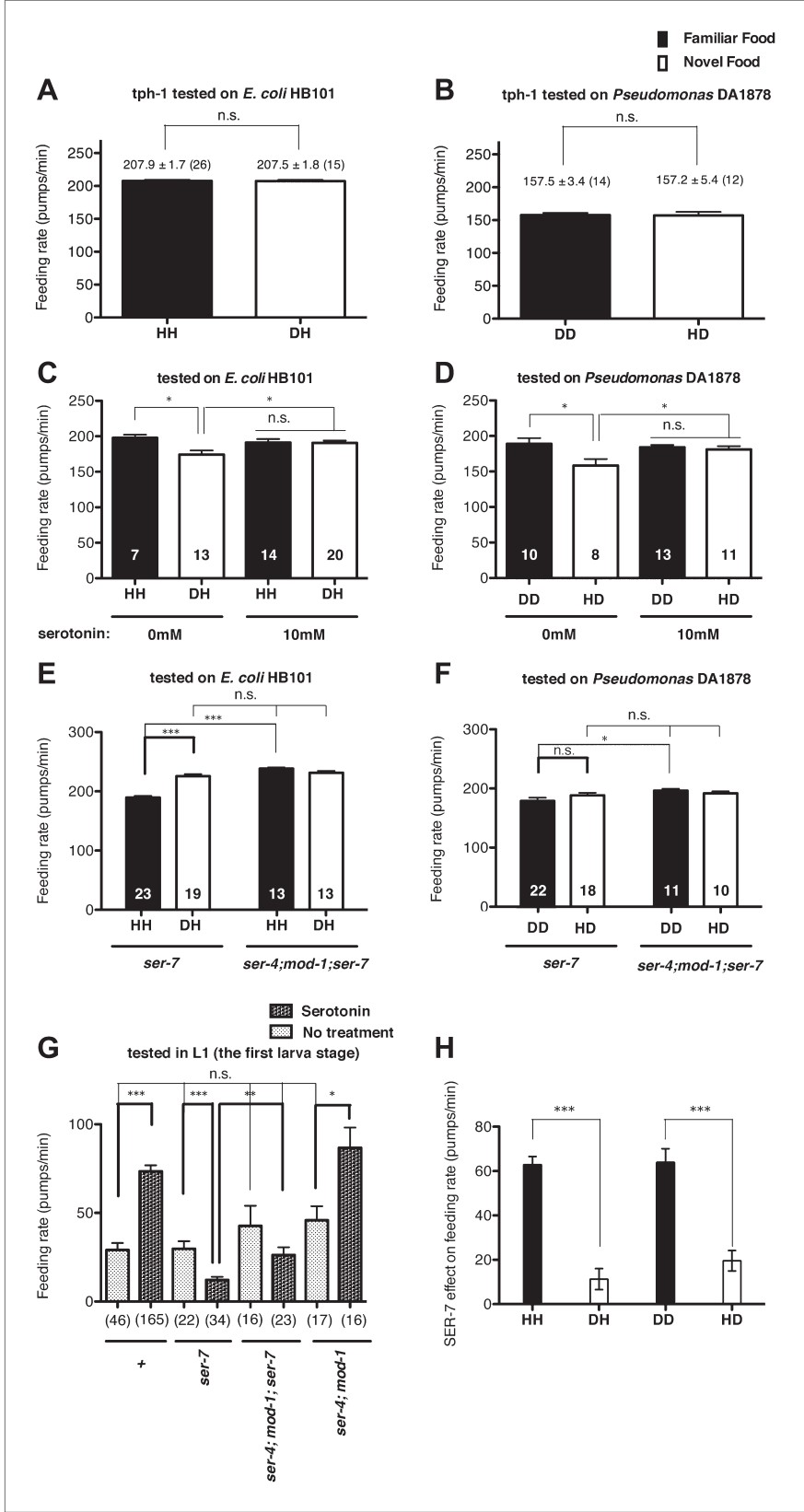

**Figure 4**. Recognition of familiar food increases feeding response by activating serotonin signaling via SER-7. SER-4 and MOD-1 are putative inhibitory receptors. (**A**)–(**B**) Familiarity of food does not alter the feeding rates in *tph-1(mg280)*. (**C**)–(**D**) Exogenous serotonin treatment selectively increases feeding rates of wild type worms on

*Figure 4. Continued on next page*

*Figure 4. Continued*

novel food to the level of the worms on familiar food. The average values of the feeding rates presented in (**C**) are 197.7 ± 4.6, 174.3 ± 5.8, 191.2 ± 4.9 and 190.7 ± 3.4 in order. The average values of the feeding rates presented in (**D**) are 189.0 ± 7.9, 158.4 ± 9.2, 184.2 ± 3.3 and 181.2 ± 4.6 in order. (**E**)–(**F**) *ser-7(tm1325)* is defective in increasing feeding response to familiar food compared to novel food. Familiarity of food does not alter the feeding rates in *ser-4(ok512); mod-1(ok103); ser-7(tm1325)*. The average values of the feeding rates presented in (**E**) are 189.7 ± 2.6, 225.5 ± 3.6, 238.3 ± 2.1 and 231.3 ± 3.4 in order. The average values of the feeding rates presented in (**F**) are 178.8 ± 5.5, 188.4 ± 4.2, 196.3 ± 3.2 and 191.9 ± 3.1 in order. (**G**) Serotonin controls feeding positively via SER-7 and negatively via SER-4 and MOD-1. The feeding rate of the *ser-4; mod-1; ser-7* triple null mutant is not altered by serotonin treatment. These assays were conducted on 3- to 5-hr-old L1 larvae, which pumped much more slowly than the adults used in other measurements. The average values of the feeding rates presented in (**G**) are 29.1 ± 4.0, 73.6 ± 3.3, 29.8 ± 4.2, 12.2 ± 1.8, 42.7 ± 11.3, 26.3 ± 4.3, 45.9 ± 8.0 and 86.8 ± 11.5 in order. (**H**) Serotonin signaling via SER-7 that activates the feeding response is more active on familiar food than novel food. The *y* axis indicates the difference in the feeding rates between wild-type and *ser-7(tm1325)* animals. Each value corresponds to the difference in the feeding rates between wild-type and the *ser-7* null mutant presented in ***Figure 7A and B***. Data shown as mean ± SEM, n.s., not significant ($p \geq 0.05$), *$p < 0.05$, **$p < 0.01$, ***$p < 0.001$; for ***Figure 4A and B***, unpaired t-test and Mann–Whitney U test (two-tailed), for ***Figure 4C–G***, one-way ANOVA, post hoc Tukey test and for ***Figure 4H***, Student's t test (two-tailed; see 'Detailed data analysis' in 'Materials and methods'). The number of animals tested (n ≥ 3 independent assays per each group) is shown in parentheses or at the bottom of each bar.

two mutant animals were not different on novel food. Interestingly, the amount of suppression by the inhibitory serotonin signal on familiar food, that is the difference between feeding rates of the *ser-4; mod-1; ser-7* mutant and the *ser-7* null mutant animals on familiar food, was greater on HB101 than DA1878 (***Figures 4E,F and 7F***). Although it is not clear why the activities of the inhibitory serotonin signal on the two bacteria are different, it explains the unexpected observation seen in the feeding rate differences between the *ser-7* null mutant animals on familiar food and novel food in ***Figure 4E and F***. We next tested whether the inhibitory serotonin signal is essential for recognition of familiar food or its regulation of feeding by comparing the feeding rates of the *ser-4; mod-1* double null mutant animals on familiar food and novel food. Consistent with the idea that SER-7 is the major serotonin receptor, the double mutant, like wild type, showed increased feeding responses on familiar food compared to novel food (***Figure 9***).

To test if serotonin feeding signaling via SER-7 indeed gets activated by recognition of familiar food, we compared the differences between feeding rates of wild-type and the *ser-7* null mutant animals (the SER-7 effect) on familiar food with the differences on novel food. Any feeding rate difference between wild-type and the *ser-7* mutant animals indicates active serotonin signaling via SER-7 because *ser-7* specifically affects serotonergic signaling, with some contribution from basal activity of SER-7 in the absence of serotonin (***Hobson et al., 2003***) (***Figure 7E***). If serotonin signaling is equally active on familiar food and novel food, we expect the SER-7 effect to be similar on familiar food and novel food. However, on familiar food the SER-7 effect was far greater than on novel food (***Figure 4H***; see 'Detailed data analysis' in 'Materials and methods'), suggesting that serotonin signaling via SER-7 is indeed more active on familiar food than novel food. We concluded that recognition of familiar food increases the feeding response mainly by activating serotonin signaling via SER-7.

## Serotonin from ADF chemosensory neurons directly activates the feeding circuit

To gain insight into how serotonin signals familiar bacteria, we asked which serotonergic neurons regulated the feeding response. Serotonin is detected in five types of neurons in *C. elegans* hermaphrodites: NSM, ADF, HSN, RIH and AIM (***Sze et al., 2000***). RIH and AIM obtain serotonin by taking up extracellular serotonin (***Jafari et al., 2011***). HSN is also unlikely to be necessary for the behavioral plasticity because feeding rates of males, which do not have HSN, were also greater on familiar food than the rates on novel food (***Figure 10***). We therefore hypothesized that either NSM or ADF uses serotonin to control feeding. The NSM neurons are a pair of secretory neurons located in the pharynx, whereas the ADF neurons are a pair of chemosensory neurons located outside the pharynx (***Sze et al., 2000***) that have been suggested to sense bacteria (***Bargmann and Horvitz, 1991***). We asked if serotonin either in ADF or in NSM regulates the feeding response by expressing *tph-1* cDNA in the *tph-1*

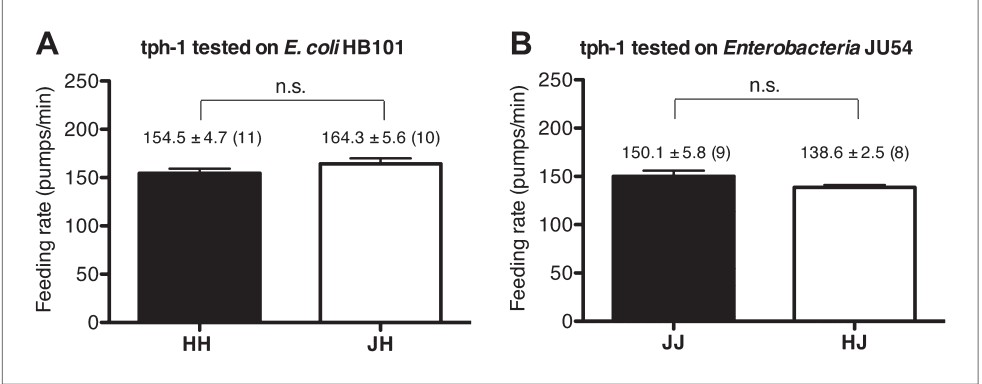

**Figure 5**. Serotonin is required to increase feeding in response to familiar food. Feeding rates of *tph-1(mg280)* on HB101 (**A**) and JU54 (**B**) after a 7- to 8-hr interval from training the animals on one or the other bacterium. Data shown as mean ± SEM, n.s., not significant (p≥0.05), unpaired t-test and Mann–Whitney U test (two-tailed). The number of animals tested (n ≥ 3 independent assays per each group) is shown in parentheses.

null mutant using either the *srh-142* promoter or the *ceh-2* promoter. The *srh-142* promoter drives expression specifically in ADF and the *ceh-2* promoter drives expression in NSM and three additional neurons (**Liang et al., 2006**). We found that restoring serotonin synthesis in ADF, but not in NSM, rescued the feeding response in the *tph-1* mutant (**Figure 8A,B**), suggesting that ADF regulates feeding in response to familiar bacteria. Laser killing of ADF, but not NSM, also decreased feeding on familiar food (**Figure 8C,D**)—the difference in feeding rates between ADF-minus and mock-operated animals (49.8 ± 7.4; feeding rates of ADF-minus and mock-operated animals were 199 ± 7.7 and 249 ± 3.5, respectively) was comparable to the difference in the feeding rates between *tph-1* and wild-type animals (58.4 ± 3.5; feeding rates of *tph-1* and wild-type animals were 207.7 ± 1.9 and 266.1 ± 3.0, respectively), further supporting the idea that ADF regulates feeding in response to familiar bacteria.

In *tph-1; Ex[ADF::tph-1(+)]* animals, in which serotonin synthesis activity was restored only in ADF, serotonin was detected in other serotonergic neurons in addition to ADF (**Figure 11B**; see 'Immunohistochemistry' in 'Materials and methods' for details). This suggests that serotonin synthesized by ADF might act in either of two possible ways: it could activate SER-7 directly, or it could be taken up and subsequently released by other serotonergic neurons. To distinguish between these possibilities, we compared the feeding rates of *tph-1; Ex[ADF::tph-1(+)]* animals with or without *mod-5*. MOD-5 is a serotonin transporter required to take up extracellular serotonin into some serotonergic neurons (**Ranganathan et al., 2001**; **Jafari et al., 2011**). In *mod-5; tph-1; Ex[ADF::tph-1(+)]*, serotonin was detected only in ADF (**Figure 11C**), suggesting that *mod-5* loss blocks serotonin uptake into other cells. If serotonin synthesized by ADF acts only through other serotonergic neurons, *mod-5* loss should substantially

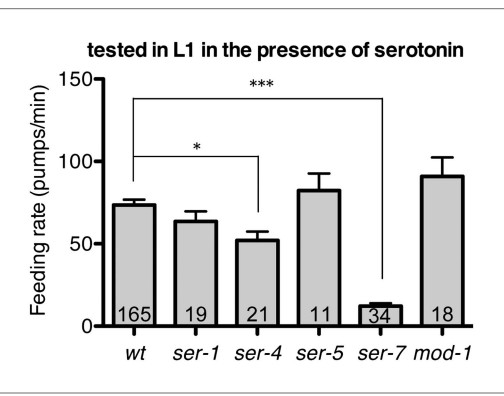

**Figure 6**. Feeding rates of wild type and five serotonin receptor null mutants in presence of serotonin. Among five serotonin receptor null mutants, only *ser-7(tm1325)* failed to activate feeding in response to serotonin. A null mutation in *ser-4* also decreased the feeding rate in presence of serotonin but the effect was relatively small. These assays were conducted on 3- to 5-hr-old L1 larvae, which pumped much more slowly than the adults used in other measurements. The average values of the feeding rates presented in this figure are 73.6 ± 3.3, 63.5 ± 6.2, 52.1 ± 5.4, 82.3 ± 10.5, 12.2 ± 1.8 and 90.9 ± 11.6 in order. *p<0.05, ***p<0.001; one-way ANOVA, post hoc Tukey test. The number of animals tested (n ≥ 2 independent assays per each group) is shown at the bottom of the bar.

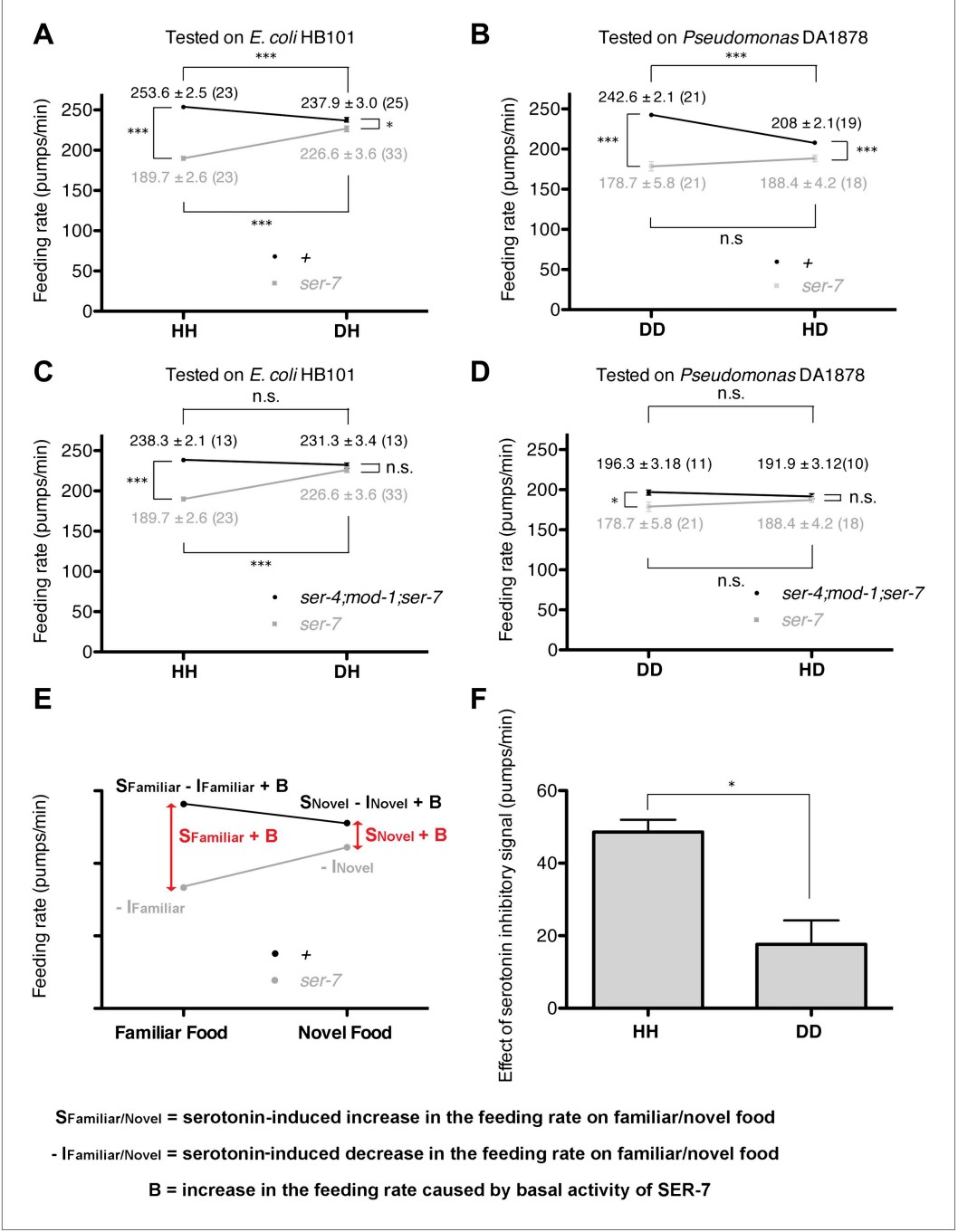

**Figure 7**. Feeding rates of wild-type, *ser-7* single and *ser-4*; *mod-1*; *ser-7* triple null mutant on HB101 and DA1878 and model of feeding regulation by serotonin. (**A**)–(**B**) Feeding rates of wild-type(+) and *ser-7(tm1325)* on HB101 (**A**) and DA1878 (**B**) after a 7- to 8-hr interval from training the animals on one or the other bacterium. Wild-type worms feed more actively on familiar food than novel food. On novel food the feeding rate of wild-type is slightly higher than that of the *ser-7* null mutant. The difference may be due to constitutive activity of SER-7; that is, SER-7 is active to some extent even in absence of its ligand, serotonin (***Hobson et al., 2003***). (**C**)–(**D**) Feeding rates of *ser-4(ok512)*; *mod-1(ok103)*; *ser-7(tm1325)* and *ser-7(tm1325)* on HB101 (**C**) and DA1878 (**D**) after a 7- to 8-hr interval from training the animals on one or the other bacterium. *ser-7(tm1325)* is defective in increasing the feeding response to familiar food compared to novel food. Familiarity of food does not alter feeding rates in *ser-4(ok512)*; *mod-1(ok103)*; *ser-7(tm1325)*. Like the positive SER-7-mediated signal, the inhibitory SER-4- and MOD-1-mediated serotonin signaling is more active on familiar food than novel food, but it decreases the feeding rate. (**E**) A simple linear model explaining how different serotonin receptors might contribute to the regulation of *Figure 7. Continued on next page*

*Figure 7. Continued*

pumping on familiar food and on novel food. There are three effects: B: Basal activity of SER-7, S: Serotonin-stimulated activity of SER-7, and −I: Serotonin-stimulated activity of inhibitory serotonin receptors SER-4 and MOD-1. The net effect of serotonin on wild-type(+) pumping is S + B − I; the net effect on pumping in a mutant lacking SER-7 is −I. While it is presented as an aid to thinking about the results, none of the results presented in the paper depend on this model. *Figure 4H*, in particular, is a direct measurement of the effect of SER-7 under differing conditions, calculated as the difference in feeding rates between wild-type(+) and the *ser-7* null mutant worms. A change in this number suggests the action of serotonin via SER-7. We use this as the measure of serotonin action via SER-7 because it is model-independent and robust. (**F**) Serotonin signaling via SER-4 and MOD-1 that suppresses the feeding response on familiar food is more active on HB101 than DA1878. The *y* axis indicates the difference in the feeding rates between *ser-4(ok512); mod-1(ok103); ser-7(tm1325)* and *ser-7(tm1325)* animals. Each value corresponds to the difference in the feeding rates between the triple null mutant and the *ser-7* null mutant presented in (**C**) and (**D**). For (**A–D**) and (**F**), data shown as mean ± SEM, n.s., not significant (p≥0.05), *p<0.05, ***p<0.001; for (**A–D**), one-way ANOVA, post hoc Tukey test and for (**F**), Student's *t* test. The number of animals tested (n ≥ 3 independent assays per group) is shown in parentheses.

decrease the feeding rate of *tph-1; Ex[ADF::tph-1(+)]*. However, we found that ADF could activate feeding as effectively in the absence of *mod-5* as in its presence (*Figure 8A*), suggesting that serotonin from ADF directly activates SER-7.

Next, we confirmed that ADF regulates feeding through SER-7. A *ser-7* null mutation suppressed the rescue effect of restoring serotonin in ADF in the *tph-1* null mutant (*Figure 8A*). To understand how serotonin increases feeding at a neural circuit level, we asked where SER-7 acts. SER-7 is expressed mostly in pharyngeal neurons (*Hobson et al., 2006*), which regulate the motions of pharyngeal muscles (*Avery and Horvitz, 1989*). Among the pharyngeal neurons, MC is particularly interesting because it is essential for normal fast feeding on bacteria (*Avery and Horvitz, 1989*), and SER-7 was suggested to activate MC (*Hobson et al., 2006*; *Song and Avery, 2012*). To ask if SER-7 acts in MC, we expressed SER-7 in the *ser-7* null mutant using the *flp-21* and the *flp-2* promoters. The *flp-21* and the *flp-2* promoters drive expression in several neurons, and the expression patterns of the two promoters overlap only in MC and M4 (*Kim and Li, 2004*). We found that both *pflp-21*::SER-7 and *pflp-2*::SER-7 fully rescued the feeding rate in the *ser-7* mutant in response to familiar food as well as serotonin (*Figure 8E,F*; Part of the data in *Figure 8E,F* were reported previously in *Song and Avery, 2012* and were re-analyzed and presented here.). In contrast, expression of SER-7 in M4 and occasionally in M2 using the *ser-7b* promoter failed to alter the pumping rate and had only a small effect on pumping in the *ser-7* null mutant in response to serotonin and familiar food, respectively (*Figure 8E,F*), suggesting that SER-7 in MC activates pharyngeal pumping. The failure of rescue is unlikely to be due to insufficient expression because expression of the rescue construct significantly activated isthmus peristalsis, the other feeding motion in *C. elegans*, which is controlled by M4 (*Song and Avery, 2012*). To test whether SER-7 indeed acts through MC, we used an *eat-2* null mutation to test if blocking neurotransmission from MC suppresses the rescue effect of *pflp-21*::SER-7 in the *ser-7* mutant. *eat-2* encodes a nicotinic acetylcholine receptor subunit specifically localized in the pharyngeal muscles postsynaptic to MC (*McKay et al., 2004*). Thus, an *eat-2* null mutation selectively blocks cholinergic transmission from MC to the pharyngeal muscles. We found that the *eat-2* null mutation suppressed the rescue effect of *pflp-21*::*ser-7(+)* in response to serotonin (*Figure 8E*), supporting our hypothesis that SER-7 in MC increases the feeding response. In summary, we conclude that serotonin released from extrapharyngeal ADF increased feeding in response to familiar bacteria mainly by activating SER-7 in MC directly, which in turn activates cholinergic transmission from MC to the pharyngeal muscles.

## Recognition of familiar bacteria activates ADF and increases serotonin release from ADF

We next asked why ADF increases feeding response on familiar but not novel bacteria. A simple explanation is that only familiar bacteria can activate ADF, and this activation causes increased serotonin release. We tested the hypothesis first by asking if ADF is more active on familiar bacteria than novel bacteria by directly measuring the response of the ADF neurons to novel and familiar bacteria using ratiometric calcium imaging. For this, we imaged the ADF neurons using the genetically-encoded

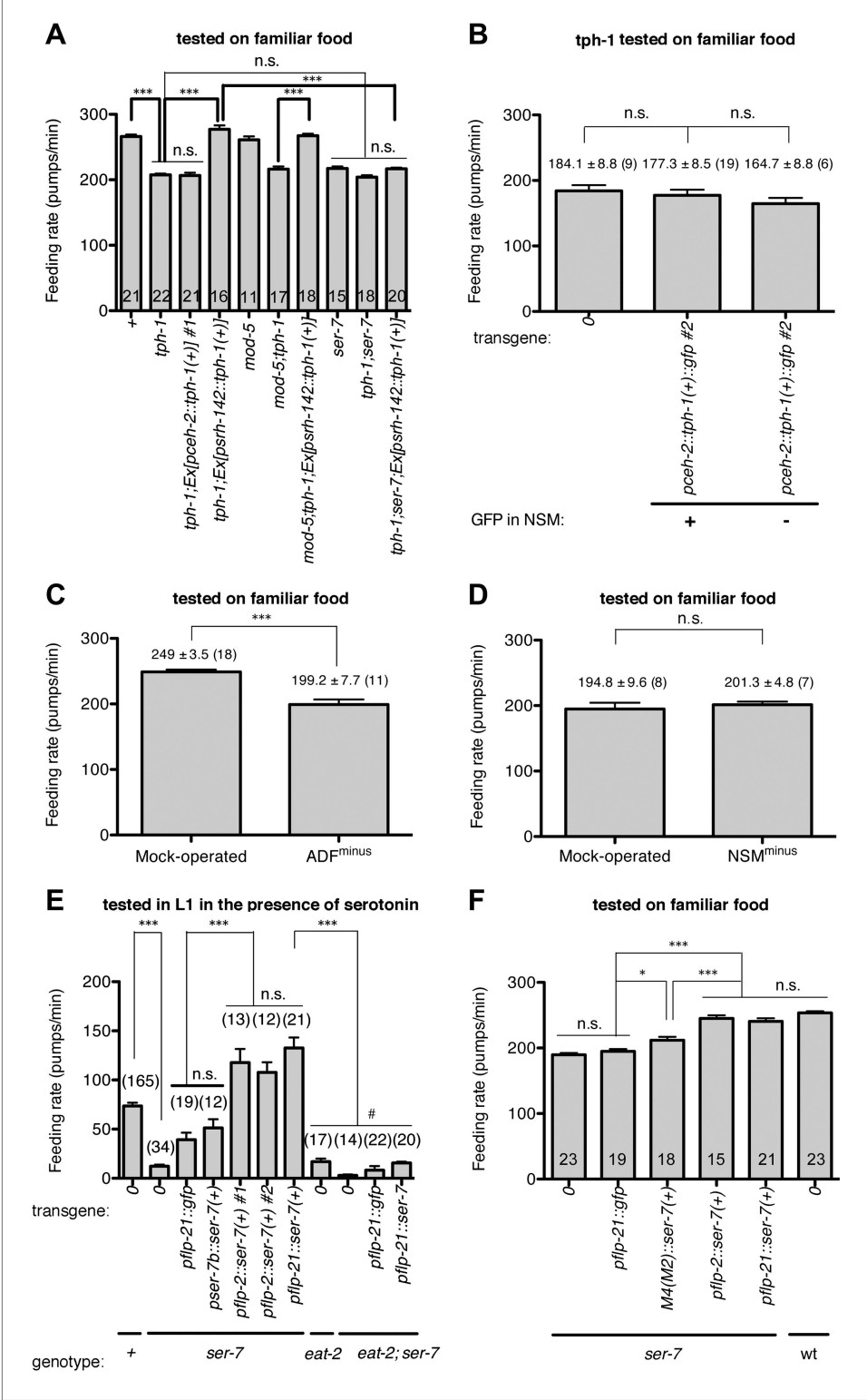

**Figure 8**. Serotonin from ADF activates feeding in response to familiar food mainly by directly activating SER-7 in MC pharyngeal motor neurons. Active SER-7 in MC (and possibly in M4) acts mainly via cholinergic transmission from MC to the pharyngeal muscles. (**A–B**) *tph-1* expression in ADF, but not in NSM, restores feeding rate in the *tph-1* null mutant. The rescue effect is suppressed by loss of *ser-7*, but not by loss of *mod-5*. No difference was found in feeding rates between the *tph-1* single null mutant, the *ser-7* single null mutant and the *tph-1; ser-7* double null mutant. The average values of the feeding rates presented in (**A**) are 266.1 ± 3.0, 207.7 ± 1.9, 206.6 ± 4.2,

*Figure 8. Continued on next page*

*Figure 8. Continued*

277.2 ± 6.0, 261.1 ± 5.3, 216.4 ± 4.0, 267.3 ± 3.1, 217.5 ± 2.8, 204.1 ± 3.0 and 216.9 ± 1.8 in order. (**C–D**) ADF-minus animals, but not NSM-minus animals, feed significantly less in response to familiar food. (**E**) Expression of *ser-7* cDNA driven either by the *flp-2* promoter or by the *flp-21* promoter (MC, M4, and other neurons) but not by the *ser-7b* promoter (M4 only) fully restored the feeding rate in the *ser-7* null mutant in response to serotonin. The rescue effect was suppressed by blocking cholinergic transmission from MC to the pharyngeal muscles. #Pharyngeal pumping rate was lower in the *eat-2; ser-7* double null mutant than the *eat-2* single null mutant (p<0.001) and the *ser-7* single null mutant (p=0.002). The difference suggests that acetylcholine marginally activates pumping in an EAT-2-independent manner and that there is residual acetylcholine release in absence of SER-7 in response to serotonin. No difference in feeding rates was found between the *eat-2; ser-7* mutant expressing *pflp-21::gfp* and the mutant expressing *pflp-21::ser-7* cDNA. The average values of the feeding rates presented in (**E**) are 73.6 ± 3.3, 12.2 ± 1.8, 39.3 ± 7.1, 51.2 ± 8.9, 117.8 ± 13.7, 107.7 ± 10.4, 132.6 ± 10.6, 16.9 ± 3.1, 2.9 ± 1.0, 8.3 ± 4.1 and 15.6 ± 1.4 in order. (**F**) Expression of *ser-7* cDNA driven either by the *flp-2* promoter or by the *flp-21* promoter fully restored the feeding rate in the *ser-7* null mutant in response to familiar food. Expression of *ser-7* cDNA in M4 (and occasionally in M2) driven by the *ser-7b* promoter also increased the feeding rate, but the effect was relatively small. The average values of the feeding rates presented in (**F**) are 189.7 ± 2.6, 194.9 ± 3.4, 212.0 ± 5.1, 245.1 ± 4.7, 240.7 ± 4.6 and 253.6 ± 2.5 in order. Data shown as mean ± SEM, n.s., not significant (p≥0.05), *p<0.05, ***p<0.001; for (**A–B**) and (**E–F**), one-way ANOVA, post hoc Tukey test, for (**C–D**), unpaired t-test and Mann–Whitney U test (two-tailed). The number of animals tested (n ≥ 3 independent assays per each group) is shown on each bar. '0' and 'wt' in this figure indicate absence of transgene and wild type, respectively.

calcium sensor Cameleon YC3.60. Changes in intracellular calcium concentration were reported as changes in the ratio of fluorescence emission at distinct wavelengths (ΔR/R) (*Nagai et al., 2004*). Slow yet substantial increases in the activity of ADF neurons, reported as an increase in ΔR/R, were observed in response to familiar food (*Figure 12A,B*). In contrast, only marginal changes in the activity were observed in response to novel food (*Figure 12A,B*). To further quantify the response observed in ADF, we measured the averaged signed area under ΔR/R for each experimental group (see 'Ca²⁺ imaging' in 'Materials and methods' for details). Consistent with our hypothesis, the increases in ADF activity were greater in response to familiar food than novel food (*Figure 12C*). The activities in response to novel food were not different from baseline (data not shown). These data indicate that familiar food, but not novel food, activates ADF neurons.

We then asked if ADF releases more serotonin in response to familiar bacteria than novel bacteria. Since direct measurement of serotonin release from ADF in response to food is challenging, we developed a method to detect released serotonin indirectly by its uptake into other serotonergic neurons. As mentioned above, in *tph-1; Ex[ADF::tph-1(+)]* animals, serotonin signal is detected in other cells in addition to ADF (*Figure 11B*; see 'Immunohistochemistry' in 'Materials and methods' for details). Since no serotonin was detected in the *tph-1* null mutant animals (*Figure 13*) and since the ADFs are the only cells capable of synthesizing serotonin in *tph-1; Ex[ADF::tph-1(+)]*, all the serotonin in these animals must have been synthesized in ADF, and its appearance in other serotonergic neurons must have occurred after release from ADF and uptake into the other neurons. In confirmation of this hypothesis, in *mod-5; tph-1; Ex[ADF::tph-1(+)]*, serotonin is detected only in ADF (*Figure 11C*). Thus, the presence of serotonin in cells other than ADF in *tph-1; Ex[ADF::tph-1(+)]* animals is an indication of serotonin release from ADF.

We thus tested if ADF releases more serotonin in response to familiar bacteria than novel bacteria by comparing the numbers of serotonin positive serotonin-uptaking cells in *tph-1; Ex[ADF::tph-1(+)]* animals on familiar bacteria and novel bacteria. As for the feeding assay, we trained the animals on HB101 or DA1878 and tested them on HB101 or DA1878 after a 7 hr interval (*Figure 11A*; see 'Immunohistochemistry' in 'Materials and methods' for details). Consistent with our hypothesis, the increase in the number of serotonin positive serotonin-uptaking cells during the 1 hr incubation on familiar food was greater than the increase on novel food (*Figure 11D*; see 'Quantification of serotonin positive neurons' and 'Detailed data analysis' in 'Materials and methods' for details). There are several other possible explanations for the greater increase on familiar food than novel food that we cannot exclude, such as increased efficiency of serotonin uptake in serotonin-uptaking cells or increased serotonin release from NSM on familiar food compared to novel food. However, considering that an increase in the efficiency of serotonin uptake is likely to result in a decrease in the level of extracellular

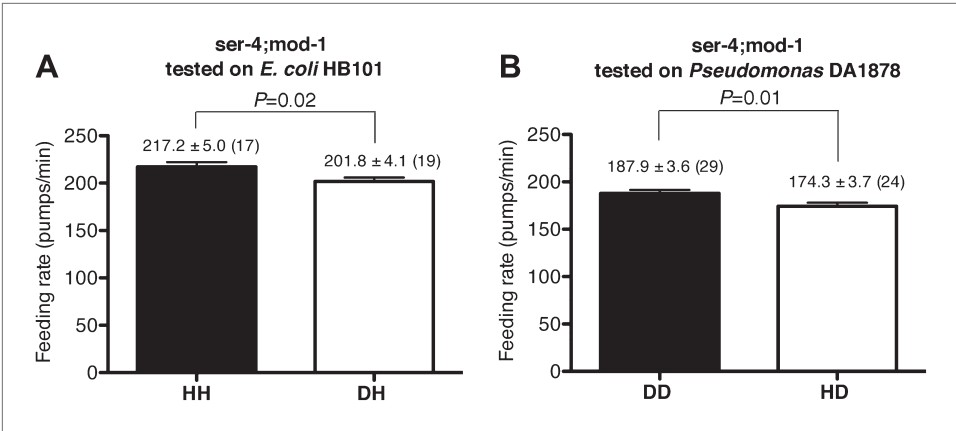

**Figure 9**. SER-4 and MOD-1 are not essential for discriminating familiar food from novel food. Feeding rates of *ser-4(ok512); mod-1(ok103)* on HB101 (**A**) and DA1878 (**B**) after a 7- to 8-hr interval from training the animals on one or the other bacterium. Like wild type worms, *ser-4(ok512); mod-1(ok103)* show increased feeding response on familiar food compared to novel food. Data shown as mean ± SEM, unpaired t-test and Mann–Whitney U test (two-tailed). The number of animals tested (n ≥ 3 independent assays per group) is shown in parentheses.

serotonin that can activate SER-7 in MC and that restoring serotonin in NSM in the *tph-1* null mutant or killing NSM in wild type did not affect the feeding response on familiar food, it is more likely that the greater increase is due to increased serotonin release from ADF. In conclusion, only familiar bacteria activate ADFs, which increases serotonin release from the neurons and subsequently activates the feeding response.

## Feeding is negatively regulated by gustatory and/or olfactory cues of novel food

Worms may recognize familiar bacteria by taste, smell or texture. To get insight into the mechanism, we tested if worms recognize familiar bacteria by their taste or smell. For this, we examined feeding rates of worms on bacteria mixed with LB broth or with medium conditioned by one of the bacteria (*Figure 14A*). The conditioned media do not contain any bacterial particles, thus, if the media alter the feeding responses to familiar food or novel food, it suggests that gustatory or olfactory cues in the media were sensed by worms and affected discrimination of familiar food from novel food. We found that the media from bacteria that are familiar to the tested worms did not alter the feeding responses to the novel bacteria (*Figure 14B,C*). In contrast, the media from novel bacteria decreased the feeding rates of worms on familiar food (HH and DD groups) to a level comparable to those of worms on novel food (DH and HD groups) (*Figure 14B,C*). These results, together with the fact that the conditioned media did not affect feeding rates on familiar food or novel food of the same bacterial type as those that conditioned the media (*Figure 14B,C*), indicate that taste and/or smell of novel bacteria overrides the stimulatory effect of familiar bacteria and suppresses feeding activation. In conclusion, worms sense taste and/or smell of novel bacteria, which negatively regulates the feeding response.

## Discussion

Exposure to a particular food plays a significant role in shaping the pattern of food intake by altering subsequent consumption of the food. Here, using the simple animal model *C. elegans*, we delineate a neural pathway by which food exposure alters later consumption of the food. We first showed that regulation of feeding by familiarity discrimination is conserved in *C. elegans* by showing that (1) Prior exposure to particular bacteria selectively increases feeding in response to those bacteria; (2) The behavior depends on the duration of exposure, but not on the timing of exposure (*Figure 3*) or nutritional status (*Figure 1B–E*); (3) *C. elegans* retains the memory of familiar bacteria for at least 7 hr (*Figure 1D,E and 2B–E*). We speculate that the *C. elegans* nervous system may have evolved this way to increase the probability of consuming wholesome food by using past food experiences as in higher vertebrates and humans. It was previously shown that naïve worms are attracted to the smell of pathogenic bacteria but develop aversion to it after experience (*Zhang et al., 2005*; *Ha et al., 2010*).

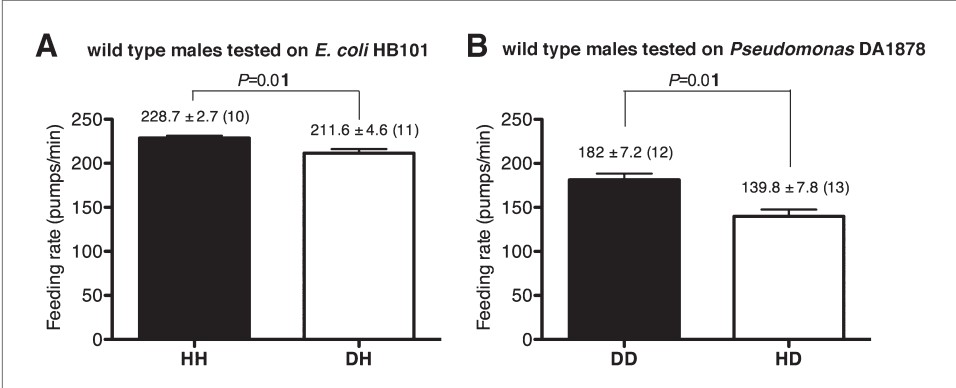

**Figure 10**. Male worms discriminate familiar food from novel food. (**A**–**B**) Feeding rates of wild-type male worms on HB101 (**A**) and DA1878 (**B**) after a 7- to 8-hr interval from training the animals on one or the other bacterium. Data shown as mean ± SEM, unpaired *t*-test and Mann–Whitney U test (two-tailed). The number of animals tested (n ≥ 3 independent assays per each group) is shown in parentheses above each bar.

The pathogenic bacteria kill worms in 4 hr (*Zhang et al., 2005*) and thus, it would be detrimental for worms to increase consumption of the bacteria. Our hypothesis can be tested by testing whether worms increase the feeding response to the pathogenic bacteria after experience.

By combining genetic analysis with imaging and immunohistochemistry, we found that recognition of familiar bacteria activates a pair of chemosensory neurons ADF, which transmits an endocrine serotonin signal that activates SER-7 in MC pharyngeal motor neurons, whose activation increases the feeding rate via cholinergic transmission from MC to the pharyngeal muscles (*Figure 15*).

## Recognition of familiar food activates feeding via an endocrine serotonin signal

Given that NSM is a prominent reservoir of serotonin in the pharynx, and that NSM is implicated in regulating the enhanced slowing response on food (*Sawin et al., 2000*), it is surprising that serotonin in NSM does not affect feeding in presence of familiar food. One plausible explanation is that NSM releases little or no serotonin, which is insufficient to activate MC neurons in the pharynx in response to familiar food. This explanation does not contradict previously reported NSM function in the enhanced slowing response (*Sawin et al., 2000*) because serotonin from NSM has only a small effect on the behavior (*Sawin et al., 2000*; *Zhang et al., 2005*). An endocrine serotonin signal from ADF, not a local serotonin signal from NSM, may have been employed for the feeding activation on familiar food to systemically control multiple behaviors and physiological adaptations. To test this possibility, it would be informative to study whether familiarity of food affects behaviors (*Horvitz et al., 1982*; *Avery and Horvitz, 1990*; *Sawin et al., 2000*; *Sze et al., 2000*) and various aspects of physiology (*Liang et al., 2006*; *Petrascheck et al., 2007*; *Srinivasan et al., 2008*) that are controlled by serotonin in presence of food (e.g., the systemic suppression of stress response that requires serotonin from ADF; *Ranganathan et al., 2001*). Further studies will be helpful to understand how recognition of familiar food contributes to survival in *C. elegans*.

## Suppressing activity of ADF or its upstream neurons by taste and smell of novel food results in selective activation of the feeding response on familiar food

How then are ADF and the downstream serotonin feeding signal controlled to increase feeding on particular bacteria after experience? Our results that conditioned media from novel bacteria override the stimulatory effect of familiar bacteria and suppress the feeding response (*Figure 14*) and that ADF is active only on familiar bacteria (*Figure 12*) indicate that perception of the smell and/or the taste of novel bacteria suppresses feeding activation on novel food by inhibiting the activity of ADF or its upstream neurons (*Figure 15*). Given that familiar food substantially increased ADF activity compared to the baseline (*Figure 12A,B*), at least two neural pathways should act antagonistically in controlling the activity of ADF or its upstream neurons. The simplest model would be that ADF or its

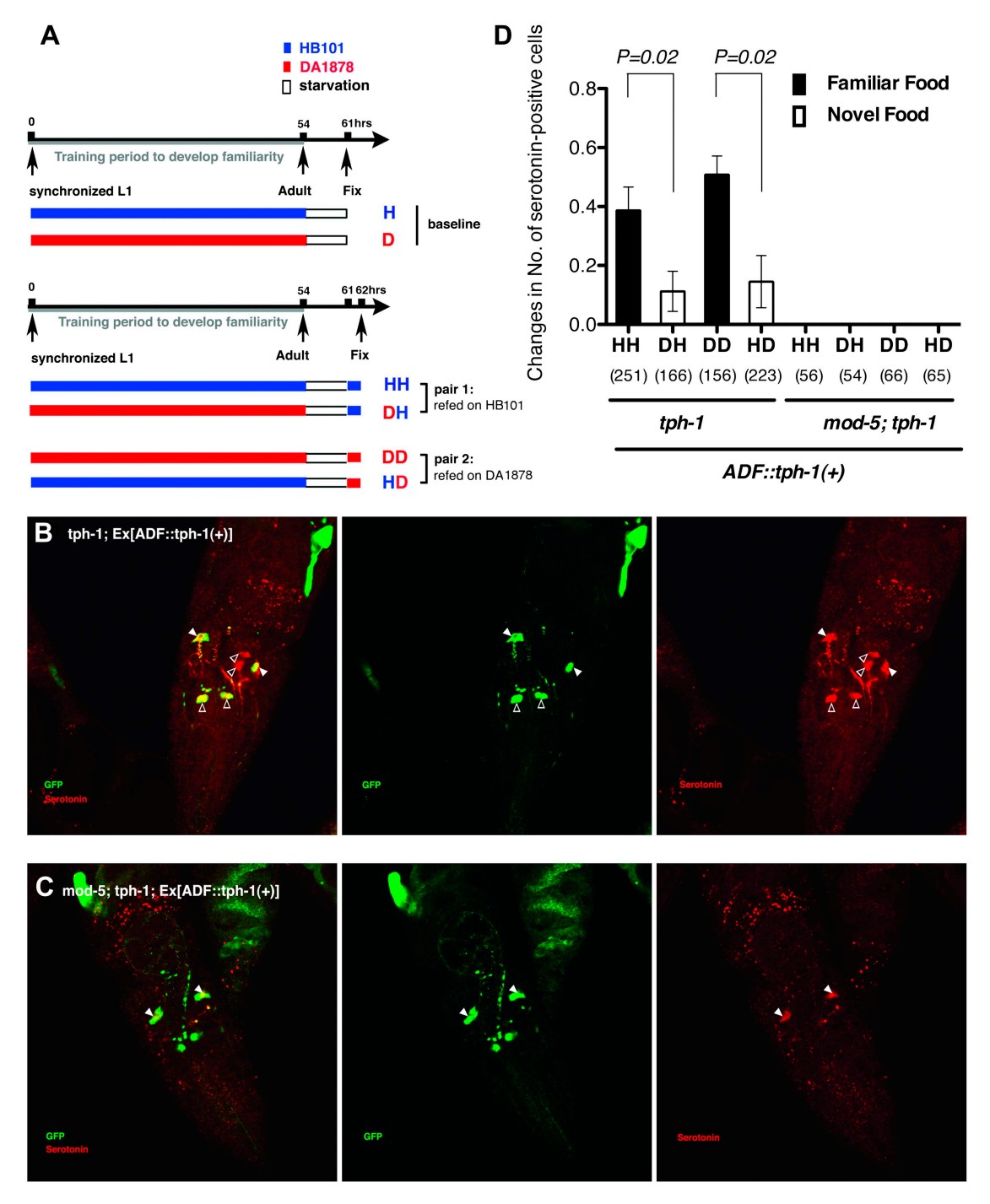

**Figure 11**. Recognition of familiar food may increase serotonin release from ADF. (**A**) Schematic of experimental design for anti-serotonin staining. Coding is as in *Figure 1A*. (**B–C**) serotonin immunoreactivity in *tph-1; Is[ptph-1::gfp]; Ex[ADF::tph-1(+)::gfp]* (**B**) and in *mod-5; tph-1; Is[ptph-1::gfp]; Ex[ADF::tph-1(+)::gfp]*, the paired control animals defective in serotonin uptake (**C**). The *Is[ptph-1::gfp]* allows the identification of NSM and ADF by GFP expression. Filled arrowheads and open arrowheads indicate ADFs and serotonin-uptaking cells, respectively. The serotonin signals not marked by arrowheads are neuronal processes. (**D**) Increase in the average number of serotonin-positive serotonin-uptaking cells during the 1 hr refeeding on familiar or novel food in *tph-1; Is[ptph-1::gfp]; Ex[ADF::tph-1(+)::gfp]* and in *mod-5; tph-1; Is[ptph-1::gfp]; Ex[ADF::tph-1(+)::gfp]* (see 'Immunohistochemistry', 'Quantification of serotonin positive neurons' and 'Detailed data analysis' in 'Materials and methods' for details). The baseline for each measurement is the average number of serotonin positive AIMs and RIH after starvation. The baseline was 2.06 ± 0.07 in animals trained on HB101 and 1.99 ± 0.05 in animals trained on DA1878. The number of animals examined (n = 3 independent assays per each group) is shown under each bar. Data shown as mean ± SEM, Student's t test (see 'Detailed data analysis' in 'Materials and methods').

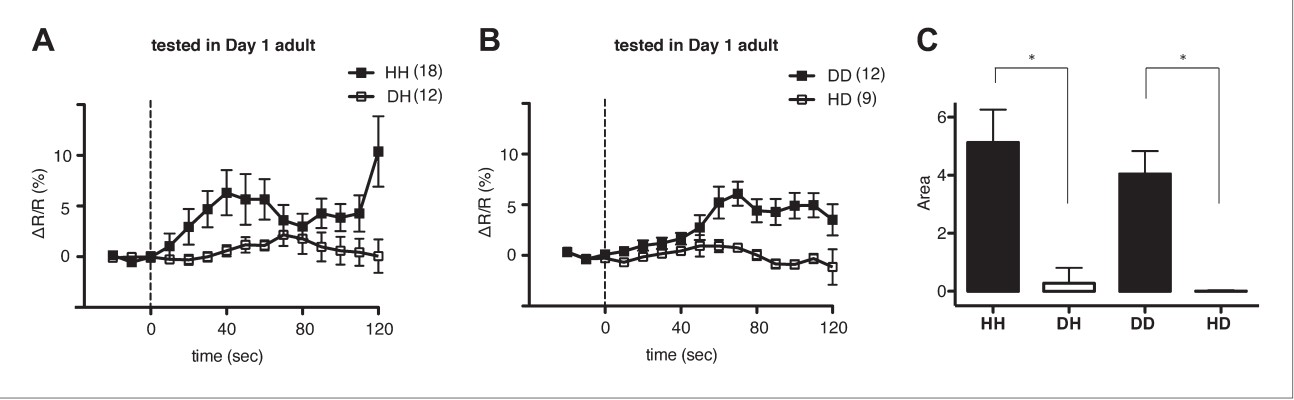

**Figure 12**. Familiar bacteria, not novel bacteria, activate ADF serotonergic neurons. (**A**–**B**). Average calcium transients in response to familiar (HH and DD groups) or novel (DH and HD groups) food. Traces represent the average percentage change from baseline over time of the fluorescence emission ratio of the ratiometric calcium sensor Cameleon YC3.60. Dashed line at $t = 0$ represents the time at which the stimulus is delivered. The number of individual recordings is indicated in parenthesis next to each group. (**C**) Average response to familiar or novel food. The bars represent the average signed area below ΔR/R between $t = 0$ and $t = 120$ s. Data shown as mean ± SEM, *$p < 0.05$; one-way ANOVA, post hoc Tukey test.

upstream neurons are positively regulated by perception of food and negatively regulated by perception of olfactory and gustatory cues of novel bacteria (*Figure 15*).

Our study predicts that novel bacteria dictate activation of ADF and the subsequent serotonin-dependent feeding response in the natural habitat where more than one bacterial type are likely to grow mixed together. Preliminary data show that *C. elegans* remembers two bacterial types at least for 24 hr (data not shown), supporting the possibility that *C. elegans* accumulates past food experience and uses them for feeding regulation in its natural habitat.

### Serotonin transmission from ADF modulates seemingly opposite experience-dependent behaviors

The following observations suggest that ADF releases serotonin and increases the feeding response when worms encounter familiar bacteria: (1) Serotonin from ADF increases the feeding response (*Figure 8A*); (2) ADF is activated selectively by familiar bacteria within 1 min (*Figure 12A,B*); (3) ADF releases more serotonin in response to familiar bacteria than novel bacteria (*Figure 11D*). Interestingly, serotonin transmission from ADF was also shown to be critical for the learned aversion to pathogenic bacteria (*Zhang et al., 2005*; *Ha et al., 2010*), which is opposite in direction to the appetitive change in feeding behavior that we describe here. For the aversive learning, it is not yet clear when the serotonin signal from ADF acts. It will be interesting to understand how serotonin signaling from ADF and the physiological context are integrated to produce seemingly opposite experience-dependent behaviors. Further studies to understand regulation of the two seemingly opposite behaviors at the neural circuit level will also help us understand how the *C. elegans* nervous system differentially encodes, maintains and retrieves the appetitive and aversive memories of bacteria.

### Conclusion

Many questions remain to be answered to fully understand the mechanism underlying recognition of familiar bacteria in *C. elegans*. How do worms sense different bacteria? What changes in the nervous system underlie the process of becoming familiar to particular bacteria during experience? Further quests to explore these unanswered questions may deepen our understanding of sensory information processing and familiarity discrimination.

### Materials and methods

#### General methods and strains

Except when stated otherwise, *C. elegans* was cultured at 19°C as described by (*Brenner, 1974*). Except in *Figure 10*, all worms used were hermaphrodites. The following mutant alleles were

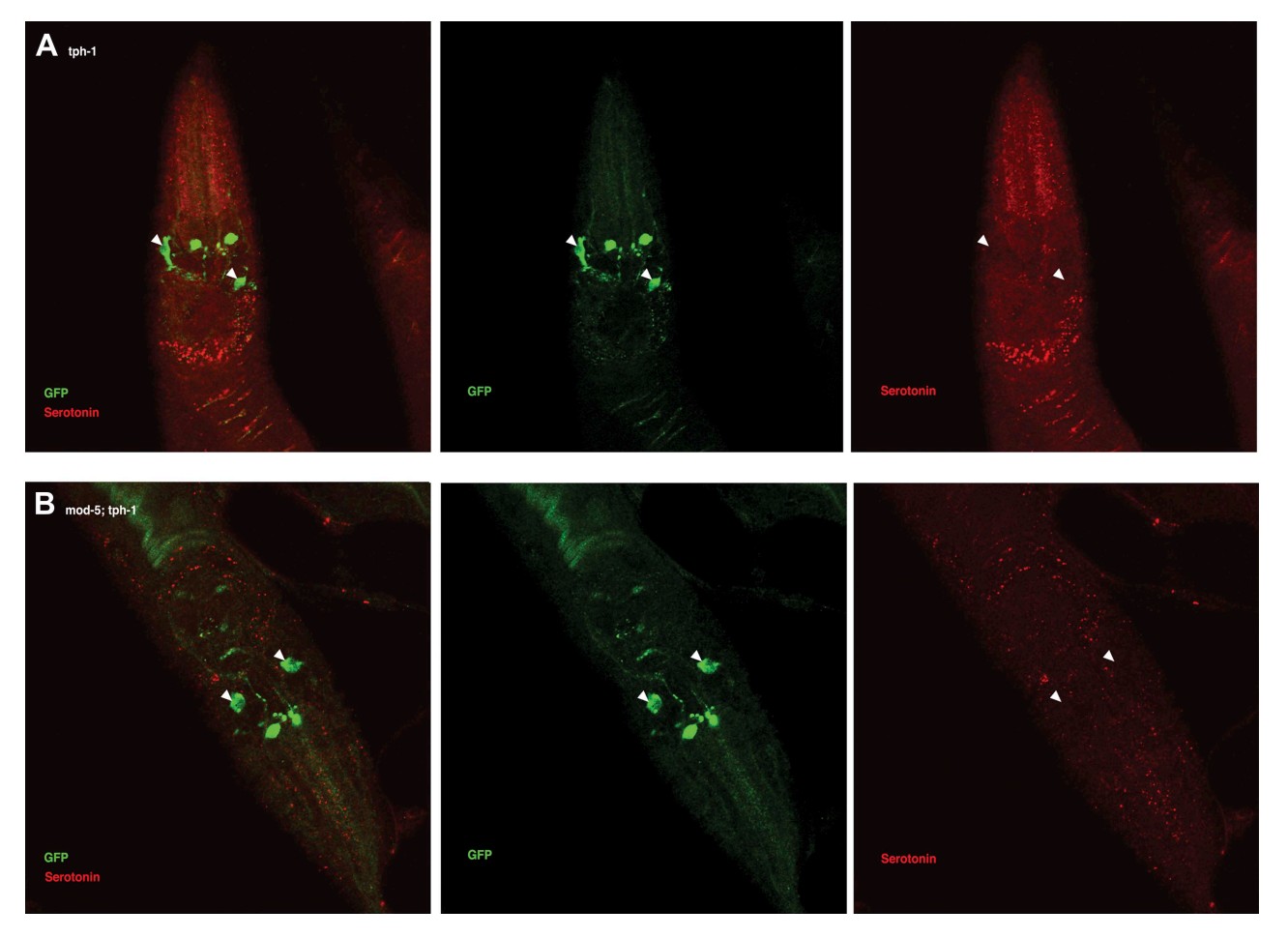

**Figure 13**. Serotonin immunoreactivity in *tph-1* single and *mod-5; tph-1* double null mutants. (**A–B**). No serotonin signal was detected in the *tph-1(mg280);Is[ptph-1::gfp]* (**A**) and in the *mod-5(n3314);tph-1(mg280);Is[ptph-1::gfp]* (**B**) mutant animals. Filled arrowheads indicate ADFs. The *Is[ptph-1::gfp]* allows the identification of NSM and ADF by GFP expression.

used: *mod-1(ok103) V, mod-5(n3314) I, ser-4(ok512) III, ser-7(tm1325) X, tph-1(mg280) II*. In the main text only the gene name is shown. The wild-type strain was N2 (*Brenner, 1974*), and the mutant strains used were DA2100: *ser-7(tm1325) X*, MT15434: *tph-1(mg280) II*, MT9772: *mod-5(n3314) I*, DA2289: *tph-1(mg280) II; kyEx947[pceh-2::tph-1(+)::gfp punc-122::gfp(+)]*, DA2290: *tph-1(mg280) II; kyEx949[psrh-142::tph-1(+)::gfp punc-122::gfp(+)]*, DA2293: *tph-1(mg280) II; ser-7(tm1325) X*, DA2294: *tph-1(mg280) II; ser-7(tm1325) X; kyEx949[psrh-142::tph-1(+)::gfp punc-122::gfp(+)]*, DA2295: *mod-5(n3314) I; tph-1(mg280) II*, DA2296: *mod-5(n3314) I; tph-1(mg280) II; kyEx949[psrh-142::tph-1(+)::gfp punc-122::gfp(+)]*, DA2301: *ser-7(tm1325) X; nyIs80[pflp-21::gfp(+)]*, DA2297: *ser-7(tm1325) X; adEx2297[pser-7::ser-7(+) pflp-21::gfp]*, DA2298: *ser-7(tm1325) X; adIs2298[pflp-21::ser-7(+) pflp-21::gfp(+)]*, DA2299: *mod-5(n3314) I; tph-1(mg280) II; yzIs71[ptph-1::gfp rol-6(su1006)]; kyEx949[psrh-142::tph-1(+)::gfp punc-122::gfp(+)]*, DA2300: *tph-1(mg280) II; yzIs71[ptph-1::gfp rol-6(su1006)]; kyEx949[psrh-142::tph-1(+)::gfp punc-122::gfp(+)]*, GR1333: *yzIs71 [tph-1::gfp, rol-6(su1006)] V*, OT180: *ser-4(ok512); mod-1(ok103) V; ser-7(tm1325) X*, DA2445: *ser-7(tm1325) X; adEx2245[pflp-2::ser-7(+) pflp-21::gfp]*, XL188: *ntIs16[ptph-1::yc3.60 lin-15(+)]*. *Pseudomonas* PA14 *pstP* and *Enterobacteria* JU54 were kind gifts from Dr. Fred Ausubel and Dr. Gary Ruvkun, respectively.

### Feeding assay

Developmentally synchronized L1 larvae were cultured until adulthood (for 54 hr at 19°C) on training food. For *Figure 1B,C*, individual animals were transferred to test food for measuring feeding rates

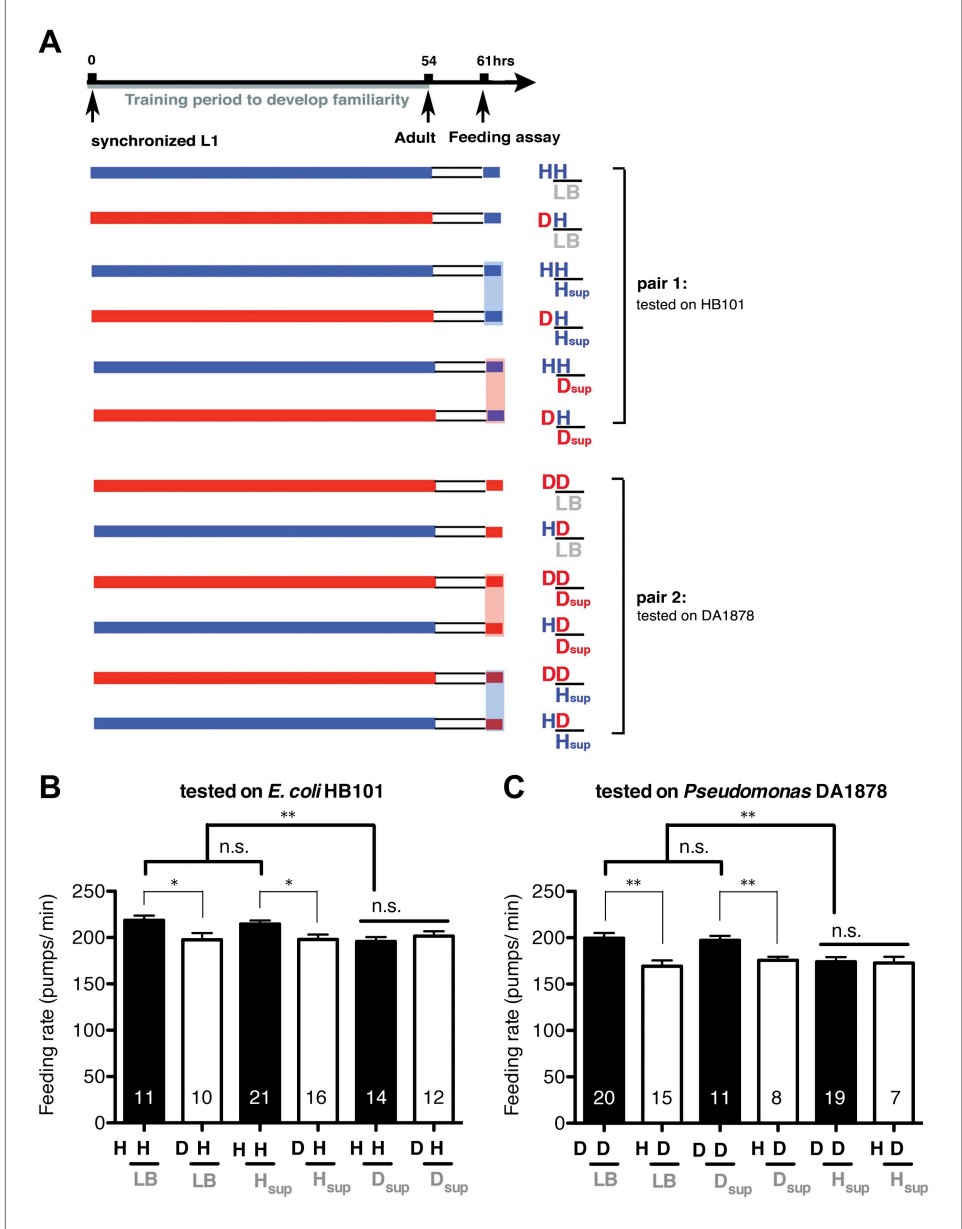

**Figure 14**. Taste and/or smell of novel bacteria override the stimulatory effect of familiar bacteria on feeding. (**A**) Experimental design for the feeding assay. Coding is as in **Figure 1A**. Each condition is coded by three symbols. The first and the second letter above the bar represent training and test food in order. The third symbol, below the bar, represents the conditioned media that was mixed with the test food. LB, $H_{sup}$ and $D_{sup}$ represent LB broth and conditioned media filtered from cultures of HB101 and DA1878, respectively. (**B–C**) Conditioned media from novel bacteria override the stimulatory effect of familiar bacteria on feeding. The average values of the feeding rates presented in (**B**) are 218.5 ± 5.0, 197.5 ± 7.3, 214.6 ± 3.9, 197.9 ± 5.4, 195.8 ± 4.7, 201.6 ± 5.2 in order. The average values of the feeding rates presented in (**C**) are 199.4 ± 5.6, 169.2 ± 6.3, 197.2 ± 4.7, 175.8 ± 3.8, 174.0 ± 5.1 and 172.8 ± 6.7 in order. Data shown as mean ± SEM, n.s., not significant ($p \geq 0.05$), *$p<0.05$, **$p<0.01$; one-way ANOVA, post hoc Tukey test. The number of animals tested ($n \geq 3$ independent assays per each group) is shown on each bar.

after removing bacteria by letting them crawl on unseeded plates for ~1 min. Removal of bacteria was checked by the absence of traces of bacteria on the track. For **Figures 1D,E, 4A–F, 5, 7, 9, 10, and 11**, individual animals were transferred to test food for measuring feeding rates after being starved for 7–8 hr on unseeded NGM plate at room temperature (RT, 23°C) after the training. We starved the animals for the following reasons: First, the 7–8 hr of starvation synchronized the nutritional status of

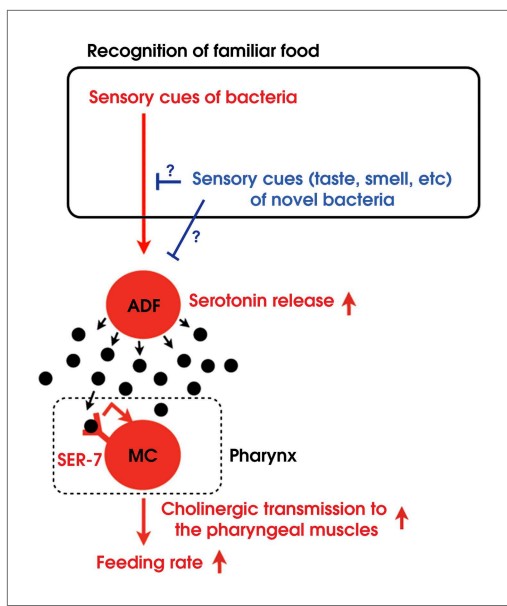

**Figure 15**. Model of activation of the feeding response by recognition of familiar food in *C. elegans*. DOI: 10.7554/eLife.00329.017

worms. Second, the 7- to 8-hr gap between the test and the last exposure to the training bacteria allows us to test if *C. elegans* can actually remember the familiar bacteria. For *Figure 3*, the animals that were cultured on HB101 or DA1878 for 54 hr at 19°C were incubated on training food or control food for either 6 or 9 hr at 19°C. Then, individual animals were transferred to test food for measuring feeding rates after 7–8 hr of starvation on unseeded NGM plates at room temperature (RT). For *Figure 8A,D,F*, the animals that were cultured on HB101 for 54 hr at 19°C (*Figure 8A,B,F*) or until adulthood (*Figure 8C,D*), were transferred to HB101 for measuring feeding rates after 7–8 hr of starvation on unseeded NGM plates at room temperature (RT). Feeding rates of individual worms were quantified by counting pharyngeal contractions 2–5 min after the transfer to test food at room temperature (about 23°C). Feeding motions of individual animals were observed with a Zeiss Stemi SV11 Apo microscope. The feeding rate of each animal (pumps per min) was calculated by averaging the three measures from each animal (pumps per 30 s) and subsequently by multiplying by 2. For each experiment, preparation of worms and reagents and feeding assays were performed in the same way in a designated place, mostly using the same batch of reagents. In contrast, conditions for feeding assays for different experiments varied in several ways, for instance, temperature and amount of test bacteria. As a result, we got consistent range of pumping rates within experiments, but not among different experiments.

## Preparation of test food and conditioned media for feeding assay

For all figures except *Figure 4A,B, 5, and 8*, test food were prepared by seeding 10 µl of bacterial culture in LB (OD = 5.0) on new NGM plates and incubating the seeded plates at RT for a defined amount of time (5.5 hr for HB101, JU54 and PA14 *pstP* and 7 hr for DA1878). For *Figure 4A,B, 5, and 8A–D*, test food was prepared in the same way except that 100 µl of bacterial culture was seeded. The variation in food preparation was necessary because *tph-1* mutants would not stay on food that was prepared from 10 µl of the culture. For conditioned media in *Figure 14*, bacterial suspensions of OD = 5.0 were prepared by collecting bacterial pellets from overnight cultures and resuspending the pellets in LB after rinsing once. After incubating the bacterial suspensions at 37°C for 12 hr while shaking, supernatants were obtained by filtering each bacterial using a 0.2-µm microfilter (Nalgene, 190-2520). 200 µl of the supernatant from each bacterial culture was added to each 4.5 ml NGM in a 35-mm plate 6.5 hr prior to the feeding assay and air-dried for 1 hr. Test food was then prepared by seeding 10 µl of bacterial culture in LB (OD = 5.0) on each plate as for other feeding assays.

## Molecular biology and generation of transgenic strains

The *ser-7b* promoter (2.2kb) for expression in M4 and M2 (*Hobson et al., 2003*) was cloned by PCR (pser-7 F: 5′- CAAACAGGTAGACAATGTTGTAAACTGTGA -3′ and pser-7 R: 5′- TTCACCCCTCAGGCTGTG -3′] from N2 genomic DNA. 1.3 kb *ser-7* cDNA (*Hobson et al., 2003*) was cloned by PCR (SER-7 cDNA F primer, 5′-*CCCGGG*ATGGCCCGTGCAGTC-3′ and SER-7 cDNA R primer 5′-*CCCGGG*CTAGACGTCACTTGGTTCGT-3′] from a cDNA pool that was reverse transcribed from N2 mRNA extracts. The *flp-21* promoter and the *flp-2* promoter were kind gifts from Dr. Chris Li (*Kim and Li, 2004*). The *ser-7b* promoter and the *flp-21* promoter were cloned into HindIII-BamHI digested pPD96.52 (Addgene plasmid 1608). *ser-7* cDNA was cloned into the EcoRI site of the two vectors containing each of the promoters. The *pflp-2::ser-7(+)* rescue construct was generated by the PCR-fusion method (*Hobert, 2002*) using the following primers: pflp-2 A primer, 5′- TCTGTGTTCACTCTACCAGGAACTTTTCTCACTTTTTAATACATATTTTCATGAAC -3′, pflp-2 A′ primer,

5'- TCTGTGTTCACTCTACCAGGA -3', pflp-2 B primer, 5'-GAGATATGTTGACTGCACGGGCC-ATGGTTTGCGACAATTGGTTTGGCAACG -3', SER-7 cDNA C primer, 5'- ATGGCCCGTGCAGTC-3', pPD9575 3'UTR D primer, 5'- GGAAACAGT TATGTTTGGTATATTGGG -3'. To generate DA2297, DA2298 and *ser-7; Ex[pflp-2::ser-7(+) pflp-21::gfp]*, germline transformation was performed in *ser-7(tm1325)* with *pser-7b::ser-7(+)* (100 ng/µl), *pflp-21::ser-7(+)* (50 ng/µl) or *pflp-2::ser-7(+)* (75 ng/µl), along with *pflp-21::gfp* (50 ng/µl) as an injection marker. For DA2298, the extrachromosomal array was integrated into the chromosome by gamma irradiation (6 krad). The integration line was out-crossed five times against DA2100. The *ceh-2* promoter and the *srh-142* promoter were kind gifts from Dr. Cori Bargmann. The *ceh-2* promoter spans 1.5 kb on chromosome I from AAGCTTAAATCTTATCAGAC to TTCTAATATTCGGAGTGAAA and the *srh-142* promoter spans 4 kb on chromosome V from TAGATTCATGTACTTGGCTC to TTTTTGCCAATATGAGTTGT.

## Laser ablation of ADF and NSM

Laser ablation of ADFs and NSMs was performed by a modified procedure (*Avery and Horvitz, 1987*). We destroyed both ADFs or NSMs in newly hatched GR1333 larvae (0–4 hr old) using a MicroPoint laser ablation system (Andor Technology USA, South Windsor, CT). For laser ablation, we mounted the larvae on 2% agarose pads containing 10 mM sodium azide. To minimize the variation caused by sodium azide, we retrieved all the mock-operated and the putative ADF-ablated animals from the agarose pad after the same incubation time (12 min). Mock-operated groups were treated in the same way except that the laser was not fired. The retrieved animals were cultured on HB101 until adulthood. After 7 hr of starvation, feeding assays were performed as described in 'Materials and methods'. Successful ablation of ADFs was confirmed by absence of the ADF or NSM GFP signals. Only data from animals in which both ADFs or NSMs were specifically destroyed are included.

## Ca²⁺ imaging

One-day-old adults were placed in a T-shaped microfluidic chamber (*Figure 16*), with the tip of their nose exposed to constantly flowing LB broth. After 30–50 s, the solution was switched to LB broth containing either DA1878 or HB101 bacteria (OD = 10.0). ADF neurons were visualized through a Zeiss plan-apochromat 63X, 1.4 NA oil immersion objective. Excitation light (436/10 nm) was provided by an X-cite 120 illuminator (Lumen Dynamics Group Inc, Ontario, Canada). For ratiometric imaging, images in cyan (480/15 nm) and yellow (535/20 nm) wavelength bands were simultaneously acquired

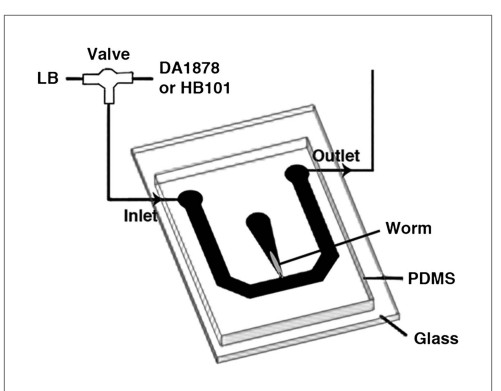

**Figure 16**. The schematic of the experimental setup for calcium imaging. The microfluidic chamber consisted of a T-shaped channel formed in PDMS bonded to a glass coverslip. The main branch of the channel was connected to an inlet and outlet, allowing LB or bacterial solutions to flow through. Switching from LB to the bacterial solutions was achieved via an upstream valve. A smaller branch orthogonal to the first one contained the worm. This tapered channel allowed immobilization of the animal while exposing only the tip of the nose to the flowing LB or bacterial solutions.

by the calcium imaging camera (Hamamatsu; ORCA-AG, Bridgewater, NJ) by means of a beam splitter (Optical Insights; OI-DV-FC, Tucson, AZ). The camera was controlled by Micromanager (*Edelstein et al., 2010*). Frames were acquired at 2–10 Hz with a 8 × 8 binning. The resulting images were analyzed off-line using custom analysis routines written in Igor Pro (Wavemetrics, Lake Oswego, OR). Briefly, fluorescence intensity I in the 480 and 535 nm wavelength images was measured in a circular region of interest (ROI) centered on the neuron. Background fluorescence I' was measured in a second ROI surrounding the first one. The raw emission ratio was computed as $R = (I_{535} - I'_{535})/(I_{480} - I'_{480}) - 0.65$, where the latter term corrects for 480-nm channel bleed-through into the 535-nm channel. This raw emission ratio was corrected for photobleaching and normalized by fitting a single exponential function to the emission ratio trace and dividing the latter by the fitted function; thus all ratio changes were expressed in terms of changes in fluorescence, ΔR/R.

To establish the time course of the ADF response to bacteria, the change in emission ratio

ΔR/R was averaged in 10 s bins for each animal, and the data then averaged across animals. The data were further quantified by measuring the signed area under ΔR/R for each animal during the first 120 s of the stimulus, and averaging across animals.

## Immunohistochemistry

Samples of DA2299 and DA2300 were prepared as for the feeding assay. Just after the 7-hr starvation we divided each group into three equal subgroups and fixed one. The remaining two groups were separately refed on either DA1878 or HB101 and fixed after 1 hr. This assay could not be done immediately after training because most of the serotonin-uptaking cells were serotonin positive in DA2300 as in wild type worms. The background was too high to detect any increase in serotonin release. Immunohistochemistry was performed using a protocol from Curtis Loer (http://home.sandiego. edu/~cloer/loerlab/anti5htlong.html) with the following antibodies: anti-serotonin rabbit IGG: S5545 (Sigma-Aldrich, Inc), Anti-GFP chicken IGG: GFP-1020 (Aves Labs, Inc), ALEXA FLOUR 488 goat anti-chicken IGG: A11039 (Invitrogen Corporation), Cy-3 conjugated donkey anti-rabbit IGG: 711-165-152 (Jackson ImmunoResearch). Failure of the co-immunostaining against serotonin and against GFP was 0%. The GFP signal was used to identify serotonergic neurons (*ptph-1::gfp*) and to find transgenics that express *tph-1* cDNA in ADF (*psrh-142::tph-1::gfp*). Due to *Is[ptph-1::gfp]* in DA2299 and DA2300, GFP signal was found in NSM, ADF, HSN and sporadically in RIH and AIM in all animals. Although *ptph-1::gfp* expresses GFP in ADF, we could still recognize the transgenic animals that express *Ex[psrh-142::tph-1::gfp]* because GFP signal in ADF is much stronger in those transgenics. The strong GFP signal in ADF was perfectly correlated with serotonin signal. In DA2300, serotonin signal was found in four different classes of serotonergic neurons (ADF, NSM, RIH and AIM). In DA2299, the control strain that is defective in serotonin uptake, serotonin signal was found only in ADF. Images were obtained with a Zeiss LSM510-meta confocal microscope using a 40× oil-immersion objective.

## Quantification of serotonin-positive neurons

To calculate the increase in the average number of serotonin-positive serotonin-uptaking cells during 1 hr of refeeding for each group (*Figure 11D*), we first blindly counted the number of serotonin positive AIMs and RIH from each animal and calculated the average number for each group. ADF, NSM and HSN were not included because serotonin was detected in all ADFs and NSMs even before refeeding and in none of HSNs even after refeeding. To minimize variation, only the animals expressing TPH-1 in both ADFs (as indicated by the presence of the strong GFP signal from *psrh-142::tph-1::gfp*) were considered for counting the number of serotonin positive AIMs and RIH. Then, we subtracted the baselines from each familiar and novel food group as follows: (Serotonin positive cell)$_{HD/HH}$ = (Average number of serotonin positive AIMs and RIH)$_{HD/HH}$ − (Average number of serotonin positive AIMs and RIH)$_{H.}$ (Serotonin positive cells)$_{DH/DD}$ = (Average number of serotonin positive AIMs and RIH)$_{DH/DD}$ − (Average number of serotonin positive AIMs and RIH)$_{D.}$ For data presentation, we combined the values from three independent experiments (see 'Detailed data analysis' in 'Materials and methods' for details).

## Assay of the effect of serotonin on feeding

To examine serotonin effects on feeding rate in absence of bacteria, feeding rates were quantified from 3- to5-hr-old L1 larvae that had never been exposed to bacteria. The feeding assay was performed with L1 larvae because it was easier to examine feeding responses of developmentally synchronized worms that are free from bacteria in large numbers. The strategy is particularly useful for the developmentally retarded mutants that carry *eat-2(ad465)*, which makes comparisons of the feeding rates in adults rather difficult. We confirmed that the effects of single null mutations of the serotonin receptors in response to serotonin are consistent in L1 and adults (data not shown) and thus, it is likely that our observations made in L1 larvae are still valid in adults. After collecting embryos by egg preparation, we incubated them on unseeded NGM plates for 2 hr. Newly hatched L1 larvae (0- to 2-hr-old) were then transferred to unseeded NGM plates and incubated for 3 hr. 15 min after mounting the larvae (3- to 5-hr-old) on 2% agarose pads containing 20 mM serotonin (H7752, Sigma-Aldrich, Inc) in M9, the feeding motions of each larva were observed using a Zeiss Axiophot microscope with a 63× objective. 2-min videos were taken from each larva with a Hitachi kP-160 CCD camera and digitized using Adobe Premiere v6.5 for quantification of feeding rates. Each experiment continued for 1 hr. Feeding rates shown in *Figures 4G and 8E* were calculated by averaging two measurements per animal (pumps per 55 s).

## Statistical analysis and data presentation

Except the data that are analysed by one-way ANOVA or by Student t test, data were statistically analysed by both the unpaired *t*-test and the Mann–Whitney U test (two-tailed). The two tests produced the same conclusions for all data analyses. For data presentation, the more conservative p value was selected. Only the familiar food group and the novel food group that were tested on the same test food were compared since the feeding rates on HB101 were significantly higher than the rate on DA1878 (in non-starved wild-type worms [p=0.006] and in 7–8 hr starved wild-type worms [p=0.003]), on JU54 (p=0.003) and since the feeding rate on PA14 *pstP* was significantly higher than the rate on HB101 (p=0.001). The *ser-7* effect on the feeding rate (shown in *Figure 4H*) for each food condition was calculated by subtracting the averaged feeding rate of the *ser-7* mutant from the rate of wild-type worms that were tested under each food condition. Student's *t*-test (two-tailed) was used to compare the *ser-7* effects on feeding rate between HH and DH groups, and between DD and HD groups (see 'Detailed data analysis' in 'Materials and methods' for details). GraphPad Prism (version 5.0) was used for statistical analysis. The effects of inhibitory serotonin signal on feeding rates in HH and DD groups (shown in *Figure 7F*) were calculated and compared as for the *ser-7* effect using the feeding rates of the *ser-7* single mutant and the *ser-4; mod-1; ser-7* triple mutant. To compare the increase in the numbers of serotonin positive cells during 1 hr of refeeding on familiar food with the increase on novel food, we tested the data shown in *Figure 11D* using Fisher's method (*Fisher, 1954*) for combining the results of several independent tests bearing upon the same overall hypothesis. We first compared the difference in the increase in the number of serotonin positive cells between HH and DH groups, and between DD and HD groups in each experiment using Student's *t* test (two-tailed). p values from three independent experiments were then combined using Fisher's method and tested by $\chi^2$ test (see 'Detailed data analysis' in 'Materials and methods' for details). For clarity, some results are presented in more than one panel. The feeding rates of wild type and the *ser-7* null mutant in presence of serotonin are shown in *Figures 4G and 6*. The feeding rates of the *ser-7* null mutant on familiar food and novel food are presented in *Figures 4E,F and 7A–D*. The feeding rates of the *ser-4; mod-1; ser-7* triple mutant on familiar food and novel food are presented in *Figures 4E,F and 7C,D*.

## Detailed data analysis

In *Figure 4H*, comparison of the differences in the feeding rates between wild-type (*N2*) and the *ser-7* null mutant animals on familiar food with the differences on novel food using Student *t*-test.

STEP 1. Calculation of the parameters (means and standard error of the means of the differences in the feeding rates between *N2* and *ser-7* under HH, DH, DD and HD) for the comparisons using Student *t*-test

$$\text{Mean}(N2_{HH}\text{–ser-7}_{HH}) = \text{Mean}(N2_{HH}) - \text{Mean}(ser\text{-}7_{HH})$$

$$\text{Standard error of the mean}(N2_{HH}\text{–ser-7}_{HH})$$
$$= [\text{Var}(N2_{HH})/n(N2_{HH}) + \text{Var}(ser\text{-}7_{HH})/n(ser\text{-}7_{HH})]^{1/2}$$

$$\text{Mean}(N2_{DH}\text{–ser-7}_{DH}) = \text{Mean}(N2_{DH}) - \text{Mean}(ser\text{-}7_{DH})$$

$$\text{Standard error of the mean}(N2_{DH}\text{–ser-7}_{DH})$$
$$= [\text{Var}(N2_{DH})/n(N2_{DH}) + \text{Var}(ser\text{-}7_{DH})/n(ser\text{-}7_{DH})]^{1/2}$$

$$\text{Mean}(N2_{DD}\text{–ser-7}_{DD}) = \text{Mean}(N2_{DD}) - \text{Mean}(ser\text{-}7_{DD})$$

$$\text{Standard error of the mean}(N2_{DD}\text{–ser-7}_{DD})$$
$$= [\text{Var}(N2_{DD})/n(N2_{DD}) + \text{Var}(ser\text{-}7_{DD})/n(ser\text{-}7_{DD})]^{1/2}$$

$$\text{Mean}(N2_{HD}\text{–ser-7}_{HD}) = \text{Mean}(N2_{HD}) - \text{Mean}(ser\text{-}7_{HD})$$

$$\text{Standard error of the mean}(N2_{HD}\text{–ser-7}_{HD})$$
$$= [\text{Var}(N2_{HD})/n(N2_{HD}) + \text{Var}(ser\text{-}7_{HD})/n(ser\text{-}7_{HD})]^{1/2}$$

$\text{Mean}(X_Y)$ is the mean of the feeding rate of animals of genotype X under Y condition; $\text{Var}(X_Y)$ is the variance of the feeding rate of animals of genotype X under Y condition; $n(X_Y)$ is number of animals of genotype X that were tested under Y condition.

STEP 2. Comparison of the differences in the feeding rates between wild-type(*N2*) and the *ser-7* null mutant animals on familiar food with the differences on novel food using Student *t*-test

A. Comparison between HH and DH

$$t = \frac{[\{Mean(N2_{HH}) - Mean(ser\text{-}7_{HH})\} - \{Mean(N2_{DH}) - Mean(ser\text{-}7_{DH})\}]}{[Var(N2_{HH})/n(N2_{HH}) + Var(ser\text{-}7_{HH})/n(ser\text{-}7_{HH}) + Var(N2_{DH})/n(N2_{DH}) + Var(ser\text{-}7_{DH})/n(ser\text{-}7_{DH})]^{1/2}}$$

p<0.001

B. Comparison between DD and HD

$$t = \frac{[\{Mean(N2_{DD}) - Mean(ser\text{-}7_{DD})\} - \{Mean(N2_{HD}) - Mean(ser\text{-}7_{HD})\}]}{[Var(N2_{DD})/n(N2_{DD}) + Var(ser\text{-}7_{DD})/n(ser\text{-}7_{DD}) + Var(N2_{HD})/n(N2_{HD}) + Var(ser\text{-}7_{HD})/n(ser\text{-}7_{HD})]^{1/2}}$$

p<0.001

These statistical analyses concluded that the difference in the feeding rates between wild-type(*N2*) and *ser-7* is greater on familiar food than the difference on novel food, suggesting that serotonin signaling via SER-7 is more active on familiar food than novel food.

In **Figure 7F**, comparison of the differences in the feeding rates between the *ser-4; mod-1; ser-7* (OT180) and the *ser-7* null mutant animals on familiar food using Student *t*-test was done in the same way.

$$t = \frac{[\{Mean(OT180_{HH}) - Mean(ser\text{-}7_{HH})\} - \{Mean(OT180_{DD}) - Mean(ser\text{-}7_{DD})\}]}{[Var(OT180_{HH})/n(OT180_{HH}) + Var(ser\text{-}7_{HH})/n(ser\text{-}7_{HH}) + Var(OT180_{DD})/n(OT180_{DD}) + Var(ser\text{-}7_{DD})/n(ser\text{-}7_{DD})]^{1/2}}$$

p<0.05

In **Figure 11D**, comparison of the numbers of serotonin positive serotonin-uptaking cells in *tph-1; Ex[ADF::tph-1(+)]* animals on familiar bacteria and novel bacteria.

STEP 1. Comparison of the increase in the average number of serotonin positive serotonin-uptaking cells in the *tph-1;Is[ptph-1::gfp];Ex[ADF::tph-1(+)::gfp]* animals that were refed on familiar food with the increase in the animals that were refed on novel food using Student *t*-test

A. Calculation of the parameters (means and standard error of the means for HH-H, DH-D, DD-D and HD-H) for the comparisons using Student *t*-test (The subtraction was for isolating the increase in the number during the 1 hr of refeeding.)

$$Mean(HH\text{-}H) = Mean(HH) - Mean(H)$$
$$Standard\ error\ of\ the\ mean(HH\text{-}H) = [Var(HH)/n_{HH} + Var(H)/n_H]^{1/2}$$

$$Mean(DH\text{-}D) = Mean(DH) - Mean(D)$$
$$Standard\ error\ of\ the\ mean(DH\text{-}D) = [Var(DH)/n_{DH} + Var(D)/n_D]^{1/2}$$

$$Mean(DD\text{-}D) = Mean(DD) - Mean(D)$$
$$Standard\ error\ of\ the\ mean(DD\text{-}D) = [Var(DD)/n_{DD} + Var(D)/n_D]^{1/2}$$

$$Mean(HD\text{-}H) = Mean(HD) - Mean(H)$$
$$Standard\ error\ of\ the\ mean(HD\text{-}H) = [Var(HD)/n_{HD} + Var(H)/n_H]^{1/2}$$

Mean(X) is mean of number of serotonin-positive serotonin-uptaking cells in group X; Var(X) is variance of number of serotonin-positive serotonin-uptaking cells in group X; $n_X$ is sample number of group X.

B.Comparison between HH-H and DH-D in each experiment

$$t = \frac{[\{Mean(HH) - Mean(H)\} - \{Mean(DH) - Mean(D)\}]}{[Var(HH)/n_{HH} + Var(H)/n_H + Var(DH)/n_{DH} + Var(D)/n_D]^{1/2}}$$

Experiment 1: p=0.239
Experiment 2: p=0.380
Experiment 3: p=0.007

C. Comparison between DD-D and HD-H in each experiment

$$t = \frac{[\{Mean(DD) - Mean(D)\} - \{Mean(HD) - Mean(H)\}]}{[Var(DD)/n_{DD} + Var(D)/n_D + Var(HD)/n_{HD} + Var(H)/n_H]^{1/2}}$$

Experiment 1: p=0.155
Experiment 2: p=0.005
Experiment 3: p=0.540

STEP 2. Comparison of the increases in number of serotonin-positive serotonin-uptaking neurons (HH-H vs DH-D and DD-D vs HD-H) using $\chi^2$ test after combining the data from three independent experiments using Fisher's method.

$$\chi^2 = -2\sum_{i=1}^{k}\log_e(p_i),$$

where $p_i$ is the p value for the $i^{th}$ hypothesis test. When the p-values tend to be small, the test statistic $\chi^2$ will be large, which suggests that the null hypotheses are not true for every test.

When all the null hypotheses are true, and the $p_i$ (or their corresponding test statistics) are independent, $\chi^2$ has a distribution with $2k$ degrees of freedom, where $k$ is the number of tests being combined. This fact can be used to determine the p value for $\chi^2$.

A. Comparison between HH and DH: $\chi^2$=14.63 (degree of freedom=6)
p=0.023
B. Comparison between DD and HD: $\chi^2$=15.40 (degree of freedom=6)
p=0.017

These statistical analyses conclude that the increases in the average number of serotonin positive serotonin-uptaking neurons of HH and DD groups are greater than the increases of DH and HD groups, respectively.

## Acknowledgements

We thank F Ausubel, C Bargmann, A Fire, R Horvitz, R Komuniecki, C Li and G Ruvkun for kindly providing us with strains and reagents. We thank the Caenorhabditis Genetic Center for strains. We thank D Raizen, C Kang, R Lin, S Robertson for their discussion and comments. Some mutations were generated by the International *C. elegans* Gene Knockout Consortium (http://www.celeganskoconsortium.omrf.org). Some mutations were generated by the National Bioresource Project for the Experimental Animal 'Nematode *C. elegans*' (http://www.shigen.nig.ac.jp/c.elegans/index.jsp).

## Additional information

### Funding

| Funder | Grant reference number | Author |
| --- | --- | --- |
| US Public Health Service | HL46154 | Leon Avery |

The funders had no role in study design, data collection and interpretation, or the decision to submit the work for publication.

### Author contributions

BS, Conception and design, Acquisition of data, Analysis and interpretation of data, Drafting or revising the article; SF, Acquisition of data, Analysis and interpretation of data, Drafting or revising the article; SL, Conception and design, Acquisition of data, Analysis and interpretation of data, Contributed unpublished essential data or reagents; LA, Conception and design, Analysis and interpretation of data, Drafting or revising the article

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
