## [Author Response]

** Not all pairwise combinations are tested in behavioral assays with HB101 always being tested against one of the others. The consistency of the behavioral results mitigates this concern but the lack of a second comparison is more problematic in the Ca++ imaging study*.

Our purpose in the Ca imaging experiment was to establish the existence of a familiarity-dependent ADF response. We achieved that, and, given the consistency of the behavioral results, we do not feel it is necessary to replicate this result for every pair of foods.

** Is it possible to elicit pharyngeal pumping by ADF activation*?

The experiment would be informative but is difficult to do. We considered doing the experiment using a transgenic strain that expresses channelrhodopsin in ADF, but we did not succeed. Exposure of tested worms to blue light is essential for neural activation using channelrhodopsin and in our set up, blue light immediately induced uncoordinated and exaggerated pharyngeal pumps in wild type worms, which were subsequently followed by complete silence of the pharyngeal muscles.

** Day-to-day variations in feeding rates are as large or larger than the familiarity effect. Depending on how the experiments were done, this could be of serious concern or simply puzzling. It should be stated explicitly whether experiments and controls for every comparison were done on the same day, using the same reagents. The variability in baseline pumping rate appears as an issue in much of the data and is of particular concern with regard to drawing negative conclusions (e.g., Figure 5a, Figure 8c–d)*.

For each comparison, worms were prepared side by side and the feeding assays were done blindly using the same reagents. To ensure repeatability, data were collected from 3–5 independent experiments.

** What were the actual values of R, i.e., of the exponential fit of the baseline I535/I480 ratio and their variations? Do ADF::gfp animals display similar rates of bleaching? The average values of the individual yellow and cyan channels should also be presented. A description of the design of the microfluidic chip (or a reference) should be given. How much time passed from picking to imaging*?Author response image 1.Changes in YFP and CFP intensity in response to familiar or novel food.The total intensity in the CFP and YFP channels was measured in a region of interest (ROI) surrounding ADF. Background intensity was subtracted from the raw traces. Traces were averaged in 10 s bins and across animals. Solutions were switched from LB broth to novel or familiar bacteria at time t = 0.

In order to rule out possible distortion due to differential rates of photobleaching or different baselines between groups, we compared the initial values for the raw emission ratio (R_0_) and the total photobleaching (as a % of initial value) during the 150 s of recording. The initial values for the raw emission ratio (R_0_) were similar for all groups: 1.10 ± 0.08, 1.17 ± 0.07, 1.18 ± 0.06, 0.98 ± 0.06 for the DA/DA, HB/HB, DA/HB and HB/DA groups respectively (mean ± SEM). In addition all groups experienced the same rates of photobleaching: 17% ± 2, 20% ± 1, 19% ± 2, 20% ± 3 for the DA/DA, HB/HB, DA/HB and HB/DA groups, respectively (mean ± SEM).

A reference for the design of the microfluidic chip is PLOS ONE 6(10):e25710. We have referenced this paper in ‘Calcium imaging’ in the Materials and methods.

One-day-old adults were placed in a T-shaped microfluidic chamber (reference above), with the tip of their nose exposed to constantly flowing LB broth. Recording was started within 1–2 minutes of placing the animals in the chamber, and the solution was switched to LB broth containing either DA1878 or HB101 bacteria (OD=10) 30-50 s after recording was initiated.

*[Editors' note: the following comments were sent to the authors upon evaluating the revised manuscript.*]

*One issue that still concerns us is the day-to-day variation issue. Of course we understand that variation happens. At the same time, the range of the day-to-day variation (about 100 pumps/min) is larger than the size of the effect (about 20). Your response mentions that you always tested worms in two conditions at the same time, and on the same day with the same reagents. Then you state that you combined data from 3–5 experiments (meaning 3–5 days presumably) together. This latter methodology makes the variability more difficult to understand. Why wouldn't the day-to-day variation average out similarly across groups? Moreover, if similar numbers of experiment- and control-animals were not assayed in each experiment, then how can data from the different days be grouped together (unless there was no day-by-day variation)? It is therefore necessary to provide more methodological details*.

Actually, the range of random day-to-day variation is quite narrow. We guess that you estimated the variation by comparing the values in different experiments. However, these differences are caused by different experimental conditions, not by random day-to-day variation. To collect data for each figure, we repeated experiments 3–5 times, and the pumping rates that were measured on different days were quite consistent as seen from the narrow distribution of pumping rates (Author response images 1 and 2, below). We attribute the consistency of the range of pumping rates within experiments to the consistency of experimental conditions.

For each experiment, preparation of worms and reagents and feeding assays were performed in the same way in a designated place, mostly using the same batch of reagents. In contrast, conditions for feeding assays for different experiments varied in several ways, for instance, temperature and amount of test bacteria. Feeding assays were performed in multiple places due to change in lab space and experimental convenience, which caused inevitable change in multiple factors including temperature for preparing worms and performing feeding assays. Additionally, feeding assays that involved the *tph-1* mutant were performed using more test bacteria (100ul of bacterial culture OD=5.0) than others (10ul) as described in ‘Preparation of test food and conditioned media for feeding assay’ in the Materials and methods. From our experience, temperature and density of bacteria significantly affect pumping rates.

We want to point out that, regardless of the variation in conditions, we consistently observed the stimulatory effect of familiar food on pumping throughout the 3–5 independent assays for each dataset (Author response image 3), which strongly supports the validity of our conclusions. We added text explaining the differences in the pumping rates among different experiments in ‘Feeding assay’ in the Materials and methods.Author response image 2.Difference in pumping rates among different datasets was not caused by day-to-day variation.These datasets were chosen as an example due to their difference in values. Data shown as mean ± SD.Author response image 3.Familiar food effect of stimulating pumping is consistently observed throughout the 3–4 independent assays for each comparison.Datasets for Figure 1 were chosen as an example. HH and DD are familiar food groups (filled circle) and DH and HD are novel food groups (open circle). HH and DH groups, and DD and HD groups that were tested each day were prepared side by side and the feeding assays were performed blindly using the same reagents. Each circle indicates an average value of 3 measured pumping rates of each worm.

*Finally, related to this issue are negative conclusions from experiments that have control pumping rates on the very low side (e.g., Figures 4b, 5a). As it is not clear that the pumping rate could be lower than such low control values, it is not clear that the conclusions stated are warranted*.

In fact, the pumping rate can be far lower than the control values (∼150 pumps/min). Blocking cholinergic transmission from MC, the motor neuron in which SER-7 acts to activate pumping in response to serotonin, to the pharyngeal muscles decreases the pumping rates of adult wild-type worms on *E*. *coli* DA837 to ∼ 30 pumps/min [Raizen, D. M., Lee, R., & Avery, L. Interacting Genes Required for Pharyngeal Excitation by Motor Neuron MC in *Caenorhabditis elegans*. *Genetics***141**, 1365-1382 (1995)]. Given that worms pump faster on *E. coli* DA837 than *E. coli* HB101, the bacterial strain that was used for the figures (unpublished observation), we believe that our conclusion that the *tph-1* mutant is defective in increasing pumping rate in response to familiar food is valid.